# Revisiting Intrinsic Reward for Exploration in Procedurally Generated Environments

**Kaixin Wang**
National University of Singapore
`kaixin96.wang@gmail.com`

**Kuangqi Zhou**
National University of Singapore
`kqzhou525@gmail.com`

**Bingyi Kang**
Sea AI Lab
`bingykang@gmail.com`

**Jiashi Feng**
ByteDance
`jshfeng@gmail.com`

**Shuicheng Yan**
Sea AI Lab
`yansc@sea.com`

## Abstract

Exploration under sparse rewards remains a key challenge in deep reinforcement learning. Recently, studying exploration in procedurally-generated environments has drawn increasing attention. Existing works generally combine *lifelong* intrinsic rewards and *episodic* intrinsic rewards to encourage exploration. Though various lifelong and episodic intrinsic rewards have been proposed, the individual contributions of the two kinds of intrinsic rewards to improving exploration are barely investigated. To bridge this gap, we disentangle these two parts and conduct ablative experiments. We consider lifelong and episodic intrinsic rewards used in prior works, and compare the performance of all lifelong-episodic combinations on the commonly used MiniGrid benchmark. Experimental results show that only using episodic intrinsic rewards can match or surpass prior state-of-the-art methods. On the other hand, only using lifelong intrinsic rewards hardly makes progress in exploration. This demonstrates that episodic intrinsic reward is more crucial than lifelong one in boosting exploration. Moreover, we find through experimental analysis that the lifelong intrinsic reward does not accurately reflect the novelty of states, which explains why it does not help much in improving exploration.

## 1 Introduction

How to encourage sufficient exploration in environments with sparse rewards is one of the most actively studied challenges in deep reinforcement learning (RL) (Bellemare et al., 2016; Pathak et al., 2017; Ostrovski et al., 2017; Burda et al., 2019a;b; Osband et al., 2019; Ecoffet et al., 2019; Badia et al., 2019). In order to learn exploration behavior that can generalize to similar environments, recent works (Raileanu & Rocktäschel, 2020; Zhang et al., 2020; Campero et al., 2021; Flet-Berliac et al., 2021; Zha et al., 2021) have paid increasing attention to procedurally-generated grid-like environments (Chevalier-Boisvert et al., 2018; Küttler et al., 2020). Among them, approaches based on intrinsic reward (Raileanu & Rocktäschel, 2020; Zhang et al., 2020) have proven to be quite effective, which combine *lifelong* and *episodic* intrinsic rewards to encourage exploration. The lifelong intrinsic reward encourages visits to the novel states that are less frequently experienced in the entire past, while the episodic intrinsic reward encourages the agent to visit states that are relatively novel within an episode. However, previous works (Raileanu & Rocktäschel, 2020; Zhang et al., 2020) mainly focus on designing lifelong intrinsic reward while considering episodic one only as a minor complement. The individual contributions of these two kinds of intrinsic rewards to improving exploration are barely investigated. To bridge this gap, we present in this work a comprehensive empirical study to reveal their contributions.

Through extensive experiments on the commonly used MiniGrid benchmark (Chevalier-Boisvert et al., 2018), we ablatively study the effect of lifelong and episodic intrinsic rewards on boosting exploration. Specifically, we use the lifelong and episodic intrinsic rewards considered in prior works (Pathak et al., 2017; Burda et al., 2019b; Raileanu & Rocktäschel, 2020; Zhang et al., 2020), and compare the performance of all lifelong-episodic combinations, in environments with and without extrinsic

rewards. Surprisingly, we find that only using episodic intrinsic rewards, the trained agent can match or surpass the performance of the one trained by using other lifelong-episodic combinations, in terms of the cumulative extrinsic rewards in a sparse reward setting and the number of explored rooms in a pure exploration setting. In contrast, only using lifelong intrinsic rewards makes little progress in exploration. Such observations suggest that the episodic intrinsic reward is the more crucial ingredient of the intrinsic reward for efficient exploration.

Furthermore, we experimentally analyze why lifelong intrinsic reward does not offer much help in improving exploration. Specifically, we find that randomly permuting the lifelong intrinsic rewards within a batch does not cause a significant drop in the performance. This demonstrates that the lifelong intrinsic reward may not accurately reflect the novelty of states, explaining its ineffectiveness.

We also compare the performance of only using episodic intrinsic rewards to other state-of-the-art methods including RIDE (Raileanu & Rocktäschel, 2020), BeBold (Zhang et al., 2020), AGAC (Flet-Berliac et al., 2021) and RAPID (Zha et al., 2021). We find simply using episodic intrinsic rewards outperforms the others by large margins.

In summary, our work makes the following contributions:

- We conduct a comprehensive study on the lifelong and episodic intrinsic rewards in exploration and find that the episodic intrinsic reward overlooked by previous works is actually the more important ingredient for efficient exploration in procedurally-generated gridworld environments.
- We analyze why lifelong intrinsic reward does not contribute much and find that it is unable to accurately reflect the novelty of states.
- We show that simply using episodic intrinsic rewards can serve as a very strong baseline outperforming current state-of-the-art methods by large margins. This finding should inspire future works on designing and rethinking the intrinsic rewards.

## 2 BACKGROUND

### 2.1 NOTATION

We consider a single agent Markov Decision Process (MDP) described by a tuple $(\mathcal{S}, \mathcal{A}, R, P, \gamma)$. $\mathcal{S}$ denotes the state space and $\mathcal{A}$ denotes the action space. At time $t$ the agent observes state $s_t \in \mathcal{S}$ and takes an action $a_t \in \mathcal{A}$ by following a policy $\pi : \mathcal{S} \to \mathcal{A}$. The environment then yields a reward signal $r_t$ according to the reward function $R(s_t, a_t)$. The next state observation $s_{t+1} \in \mathcal{S}$ is sampled according to the transition distribution function $P(s_{t+1}|s_t, a_t)$. The goal of the agent is to learn an optimal policy $\pi^*$ that maximizes the expected cumulative reward:

$$\pi^* = \arg\max_{\pi \in \Pi} \mathbb{E}_{\pi, P} \sum_{t=0}^{\infty} \gamma^t r_t \tag{1}$$

where $\Pi$ denotes the policy space and $\gamma \in [0, 1)$ is the discount factor.

Following previous curiosity-driven approaches (Pathak et al., 2017; Burda et al., 2019b; Raileanu & Rocktäschel, 2020; Zhang et al., 2020), we consider that at each timestep the agent receives some intrinsic reward $r_t^i$ in addition to the extrinsic reward $r_t^e$. The intrinsic reward is designed to capture the agent's curiosity about the states, via quantifying how different the states are compared to those already visited. $r_t^i$ can be computed for any transition tuple $(s_t, a_t, s_{t+1})$. The agent's goal then becomes maximizing the weighted sum of intrinsic and extrinsic rewards, *i.e.*,

$$r_t = r_t^e + \beta r_t^i \tag{2}$$

where $\beta$ is a hyperparameter for balancing the two rewards. We may drop the subscript $t$ to refer to the corresponding reward as a whole rather than the reward at timestep $t$.

### 2.2 TWO PARTS OF THE INTRINSIC REWARD

For procedurally-generated environments, recent works (Raileanu & Rocktäschel, 2020; Zhang et al., 2020) form the intrinsic reward as the multiplication of two parts:

$$r_t^i = r_t^{\text{lifelong}} \cdot r_t^{\text{episodic}}, \tag{3}$$

a lifelong intrinsic reward $r^{\text{lifelong}}$ that gradually discourages visits to states that have been visited many times across episodes, and an episodic intrinsic reward $r^{\text{episodic}}$ that discourages revisiting the same states within an episode. $r^{\text{lifelong}}$ is learned and updated throughout the whole training process and $r^{\text{episodic}}$ is reset at the beginning of each episode.

However, these recent works (Raileanu & Rocktäschel, 2020; Zhang et al., 2020) mainly focus on designing better lifelong intrinsic reward while considering the episodic one only as a part of minor importance. In this work, we find that the episodic intrinsic reward is actually more essential for encouraging exploration. For experiments and discussions in the following sections, we consider 4 choices of $r^{\text{lifelong}}$ and 2 choices of $r^{\text{episodic}}$ used in prior works, and introduce them below.

**ICM** Pathak et al. (2017) introduce a lifelong intrinsic reward based on a learned state representation $f_{\text{emb}}(s)$. Such representation ignores the aspects of the environment that have little impact upon the agent, and is obtained by jointly training a forward and an inverse dynamic model. The learned model makes a prediction of the next state representation $\hat{f}_{\text{emb}}(s_{t+1})$, and large prediction error implies that the agent might not explore very well. For a transition tuple $(s_t, a_t, s_{t+1})$, $r_t^{\text{lifelong}}$ is:

$$r_t^{\text{ICM}} = \frac{1}{2}\|\hat{f}_{\text{emb}}(s_{t+1}) - f_{\text{emb}}(s_{t+1})\|_2. \tag{4}$$

**RIDE** Built upon ICM, Raileanu & Rocktäschel (2020) propose a novel intrinsic reward which encourages the agent to take actions that have a large impact on the state representation. Similarly, they train inverse and forward dynamic models. The difference is that, for a transition tuple $(s_t, a_t, s_{t+1})$, $r_t^{\text{lifelong}}$ is now computed as the change in state representation:

$$r_t^{\text{RIDE}} = \|f_{\text{emb}}(s_t) - f_{\text{emb}}(s_{t+1})\|_2. \tag{5}$$

**RND** Burda et al. (2019b) propose to train a state embedding network $f(s)$ to predict the output of another state embedding network $\hat{f}(s)$ with fixed random initialization. The prediction error is expected to be higher for novel states dissimilar to the ones $f(s)$ has been trained on. Therefore, for a transition tuple $(s_t, a_t, s_{t+1})$, the prediction error is used as $r_t^{\text{lifelong}}$:

$$r_t^{\text{RND}} = \|f(s_{t+1}) - \hat{f}(s_{t+1})\|_2. \tag{6}$$

**BeBold** Built upon RND, Zhang et al. (2020) give a lifelong intrinsic reward that encourages the agent to explore beyond the boundary of explored regions. Intuitively, if the agent takes a step from an explored state to an unexplored state, then the RND error of the unexplored state will likely to be larger than that of the explored state. For a transition tuple $(s_t, a_t, s_{t+1})$, $r_t^{\text{lifelong}}$ is computed as the clipped difference of RND prediction error between $s_t$ and $s_{t+1}$:

$$r_t^{\text{BeBold}} = \max(0, \|f(s_{t+1}) - \hat{f}(s_{t+1})\|_2 - \|f(s_t) - \hat{f}(s_t)\|_2). \tag{7}$$

**Episodic visitation count** Raileanu & Rocktäschel (2020) discount their proposed lifelong intrinsic reward with episodic state visitation counts. Concretely, for a transition tuple $(s_t, a_t, s_{t+1})$, $r_t^{\text{episodic}}$ is computed as

$$r_t^{\text{ep\_count}} = \frac{1}{\sqrt{N_{ep}(s_{t+1})}} \tag{8}$$

where $N_{ep}(s)$ denotes the number of visits to state $s$ in the current episode. $N_{ep}(s)$ is post-factum computed such that the denominator is always larger than 0.

**Episodic first visits** Compared with the above episodic intrinsic reward, Zhang et al. (2020) take a more aggressive strategy: the agent only receives lifelong intrinsic reward when it visits the state for the first time in an episode. That is, for a transition tuple $(s_t, a_t, s_{t+1})$, $r_t^{\text{episodic}}$ is computed as

$$r_t^{\text{ep\_visit}} = \mathbb{1}(N_{ep}(s_{t+1}) = 1), \tag{9}$$

where $\mathbb{1}(\cdot)$ denotes an indicator function.

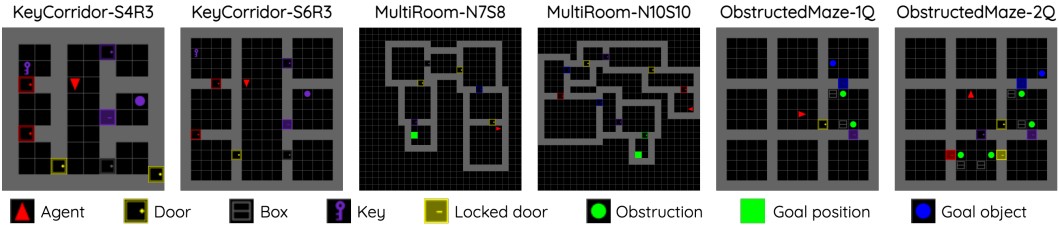

Figure 1: The 6 hard-exploration environments in MiniGrid used for our experiments.

# 3 DISENTANGLING THE INTRINSIC REWARD

As discussed above, the intrinsic reward for procedurally-generated environments can be disentangled into a lifelong term and an episodic term. In this section, we conduct extensive experiments on MiniGrid benchmark (Chevalier-Boisvert et al., 2018) to study their individual contributions to improving exploration. To this end, we ablatively compare different lifelong-episodic combinations under two settings (Sec. 3.2): (i) a sparse extrinsic reward setting where reward is only obtained upon accomplishing the task, and (ii) a pure exploration setting without extrinsic rewards. We find that episodic intrinsic reward is more important for efficient exploration, while lifelong intrinsic reward does not help much. Then, in Sec. 3.3, we also compare the approach of using only episodic intrinsic reward to other state-of-the-art methods. Finally, we analyze the reason why the lifelong intrinsic reward does not help improve exploration.

## 3.1 SETUP

**Environments** Following previous works (Raileanu & Rocktäschel, 2020; Zhang et al., 2020; Campero et al., 2021; Zha et al., 2021; Flet-Berliac et al., 2021) on exploration in procedurally-generated gridworld environments, we use the MiniGrid benchmark (Chevalier-Boisvert et al., 2018), which runs fast and hence is suitable for large-scale experiments. As shown in Fig. 1, we use 3 commonly used hard-exploration tasks in MiniGrid:

- `KeyCorridor`: The task is to pick up an object hidden behind a locked door. The agent has to explore different rooms to find the key for opening the locked door.
- `MultiRoom`: The task is to navigate from the first room, through a sequence of rooms connected by doors, to a goal in the last room. The agent must open the door to enter the next room.
- `ObstructedMaze`: The task is to pick up an object hidden behind a locked door. The key is now hidden in a box. The agent has to find the box and open it to get the key. Besides, the agent needs to move the obstruction in front of the locked door.

The suffixes denote different configurations (*e.g.*, `N7S8` for `MultiRoom` refers to 7 rooms of size no larger than 8). For each task, we choose 2 representative configurations from those used in previous works, in order to reduce the computational burden. Following prior works (Raileanu & Rocktäschel, 2020; Zhang et al., 2020), we use the partial observation of the grid. The agent chooses from 7 actions (see Appx. A.1 for details). In our sparse reward setting, the agent obtains a positive reward only when it reaches the goal position or picks up the goal object. Accomplishing the tasks requires sufficient exploration and completion of some nontrivial subroutines (*e.g.*, pick up keys, open doors). More details about the environments and tasks are deferred to Appx. A.1.

**Training details** For RL learning algorithm, we find that IMPALA (Espeholt et al., 2018; Küttler et al., 2019) used in previous works takes too long to train and requires a lot of CPU resources. Therefore, to make the computational cost manageable, we use Proximal Policy Optimization (PPO) (Schulman et al., 2017) to train the policy, which is also a high-performing actor-critic algorithm but easier and more efficient to run on a single machine than IMPALA.

Following previous works (Raileanu & Rocktäschel, 2020; Zhang et al., 2020; Campero et al., 2021; Flet-Berliac et al., 2021), we use convolutional neural networks to process the input observation (see Appx. A.2 for network architectures). The neural networks are optimized with Adam (Kingma

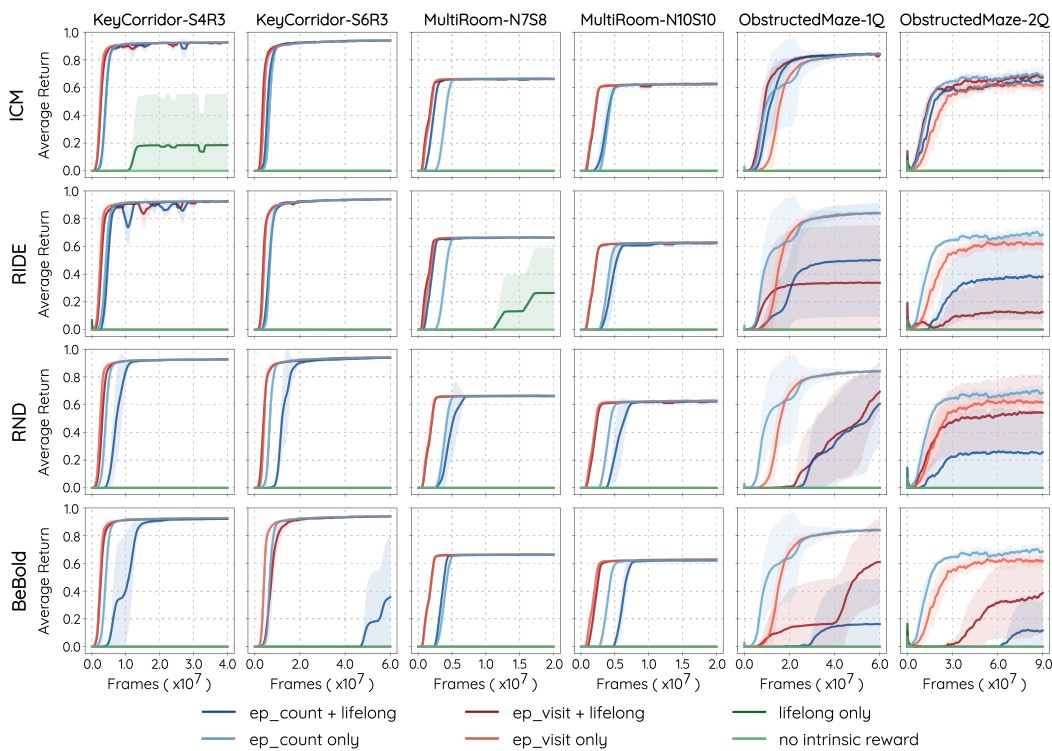

Figure 2: Performance of different combinations of lifelong and episodic intrinsic rewards on 6 hard exploration environments in MiniGrid. Note that simply using episodic intrinsic reward can achieve top performance among all combinations. Best viewed in color.

& Ba, 2015). The number of training frames are summarized in Fig. 11, which are kept the same as in (Raileanu & Rocktäschel, 2020; Zhang et al., 2020). All of our experiments can be run on a single machine with 8 CPUs and a Titan X GPU. Details about the computation cost are included in Appx. A.3.

**Hyperparameters and evaluation**   One critical hyperparameter in intrinsic-reward-based approaches is the coefficient $\beta$ that balances intrinsic and extrinsic rewards (see Eqn. 2). Varying the value of $\beta$ will have a large impact on the performance. Thus for fair comparison, we search $\beta$ for each experiment and present the results with best $\beta$. The searched values of $\beta$ as well as other hyperparameters are summarized in Appx. A.5. As in prior works, we use training curves for performance comparison. The training curves are averaged over 5 runs and the standard deviations are plotted as shaded areas. The curves are smoothed following the procedure used by Raileanu & Rocktäschel (2020). We also include the testing results in Appx. A.6.1.

## 3.2   Individual contributions of episodic and lifelong intrinsic rewards

In this section, we ablatively study how episodic and lifelong intrinsic rewards improve exploration with and without extrinsic reward (*i.e.*, goal-reaching and pure exploration). On the MiniGrid environments, we compare the performance of 15 lifelong-episodic combinations: $\{r^{\text{ICM}}, r^{\text{RIDE}}, r^{\text{RND}}, r^{\text{BeBold}}, \text{None}\} \times \{r^{\text{ep\_count}}, r^{\text{ep\_visit}}, \text{None}\}$. Here "None" refers to not using lifelong or episodic intrinsic reward. If both are not used, then it reduces to vanilla PPO without intrinsic rewards. If only one is not used, then the reward reduces to 1. For the goal-reaching task, the extrinsic reward is sparse and only provided when the agent achieves the goal. Exploration ability is measured by how fast the agent can achieve high average return on 6 environments introduced in Sec. 3.1. For the pure exploration setting, we train agents on `MultiRoom` environments without providing any extrinsic reward when agents reach the goal. `MultiRoom` environments consist of several consecutive rooms connected by doors and the agent starts from the first room (see Fig. 1). To explore the environment,

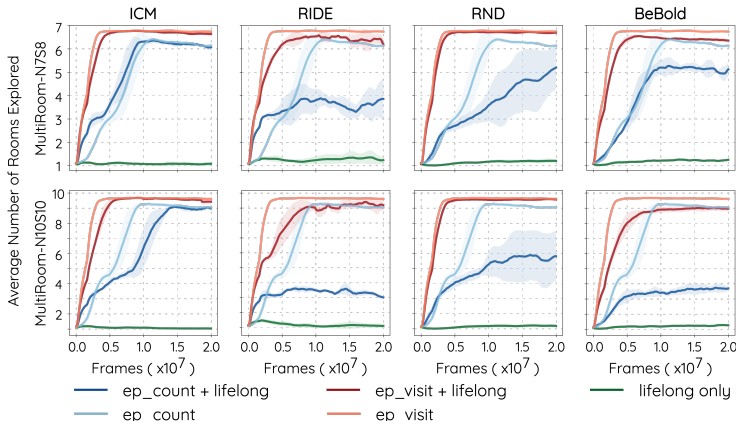

Figure 3: Average number of explored rooms without extrinsic rewards. Best viewed in color.

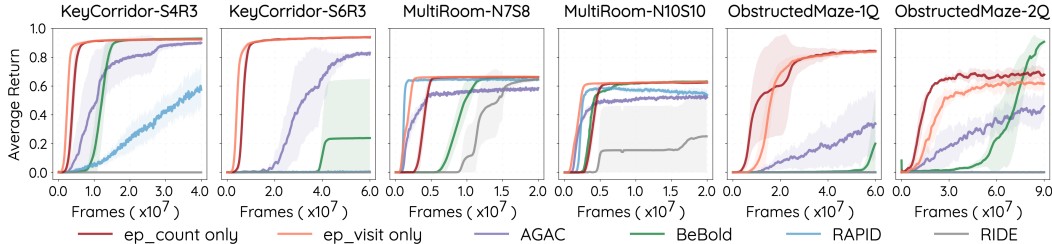

Figure 4: Comparison of only using episodic intrinsic rewards and state-of-the-art methods. Note that the result of BeBold in `ObstructedMaze-2Q` is directly taken from the associated paper (Zhang et al., 2020).

the agent needs to open each door and navigate to different rooms. Thus, we can easily use the average number of explored rooms within an episode as a proxy for quantifying the exploration ability. Here we use `MultiRoom-N7S8` and `MultiRoom-N10S10` which have 7 and 10 rooms respectively, and consider a room is explored if the agent visits any tile within this room (excluding the connecting door).

Fig. 2 shows the results of exploration with extrinsic rewards. We can see that only using episodic intrinsic reward can match or surpass the performance of combining lifelong and episodic intrinsic rewards. In contrast, only using lifelong intrinsic reward hardly makes any progress. These observations show that the episodic intrinsic reward is the key ingredient for boosting exploration. Moreover, comparing the performance of different lifelong intrinsic rewards under the same episodic intrinsic reward (i.e., dark red or dark blue curves in each column of Fig. 2), we find that recently proposed ones ($r^{\text{RIDE}}$ and $r^{\text{BeBold}}$) do not exhibit clear advantages over previous ones ($r^{\text{ICM}}$ and $r^{\text{RND}}$). Additionally, when comparing two episodic intrinsic rewards, we can see they perform comparably well for most environments. An exception is the `MultiRoom` environments, where $r^{\text{ep-visit}}$ performs slightly better. The reason might be that $r^{\text{ep-visit}}$ is more aggressive than $r^{\text{ep-count}}$ in discouraging revisits to visited states within an episode, pushing the agent to explore new rooms more quickly.

The results for the pure exploration setting are summarized in Fig. 3. Again, when only using lifelong intrinsic rewards (i.e., green curves), the agent fails to explore farther than the first 2 rooms. In contrast, it can quickly explore more rooms when augmented with episodic intrinsic rewards. This further demonstrates that episodic intrinsic reward is more important than lifelong one for efficient exploration. Moreoever, comparing $r^{\text{ep-visit}}$ and $r^{\text{ep-count}}$, we can see $r^{\text{ep-visit}}$ helps the agent explore faster. This difference in exploration efficiency explains the performance difference between $r^{\text{ep-visit}}$ and $r^{\text{ep-count}}$ in `MultiRoom` environments with extrinsic reward (see Fig. 2).

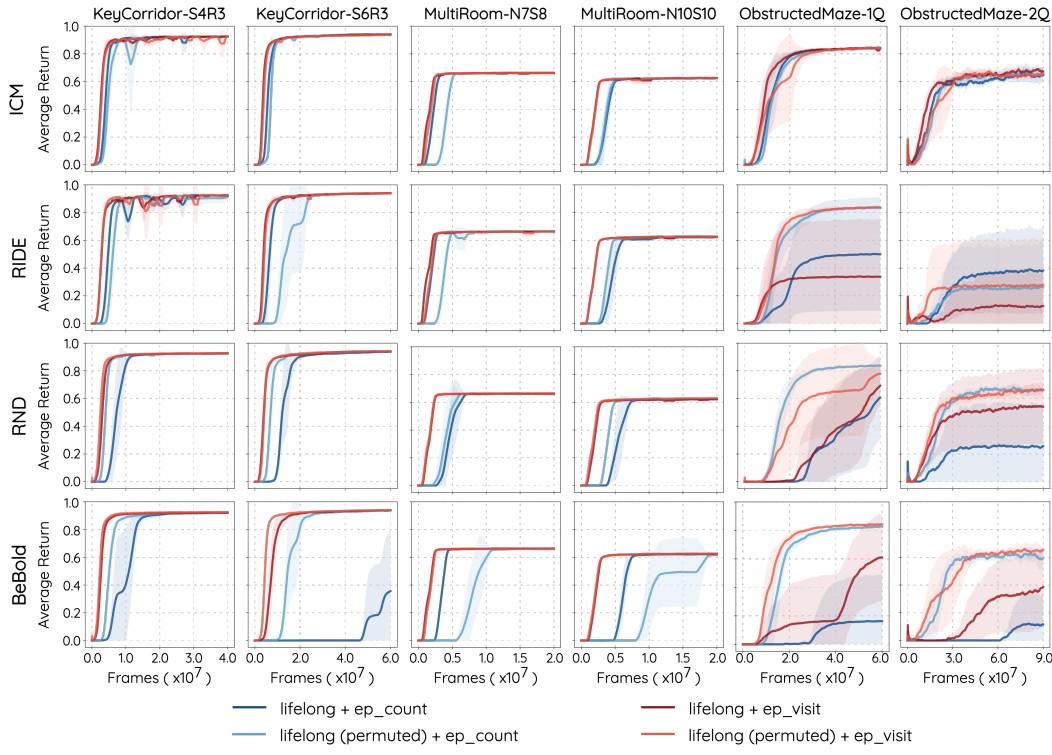

Figure 5: Performance of whether randomly permuting lifelong intrinsic rewards on 6 hard exploration environments in MiniGrid. Best viewed in color.

### 3.3 COMPARISON WITH STATE-OF-THE-ART METHODS

From the previous section, we can see that only using episodic intrinsic rewards can achieve very competitive performance. In this section, we compare its performance with other state-of-the-art approaches, including RIDE (Raileanu & Rocktäschel, 2020), BeBold (Zhang et al., 2020), RAPID (Zha et al., 2021) and AGAC (Flet-Berliac et al., 2021), under the sparse extrinsic reward setting. Among the 6 environments used, 1) for environments also used in these works, we contacted the authors and obtained the original results reported in their papers; 2) for other environments, we train agents using their official implementations and the same hyperparameters reported in their papers. Please see Appx. A.4 for more details.

As Fig. 4 shows, only using $r^{\text{ep-visit}}$ or $r^{\text{ep-count}}$ clearly outperforms other state-of-the-art methods, achieving high average return with fewer training frames. Specifically, in `KeyCorridor-S6R3`, `ObstructedMaze-1Q` and `ObstructedMaze-2Q`, it is almost an order of magnitude more efficient than previous methods. Note that the performance of BeBold in `ObstructedMaze-2Q` is directly taken from their paper but we cannot reproduce it. Compared to these sophisticated methods, simply using episodic intrinsic rewards serves as a very strong baseline. In Appx. A.6.2, we also include the comparison results on other environments in MiniGrid benchmark.

### 3.4 WHY DOESN'T LIFELONG INTRINSIC REWARD HELP?

As shown in Sec. 3.2, lifelong intrinsic reward contributes little to improving exploration. In this section, we make preliminary attempts to analyze the reason for its ineffectiveness.

To begin with, we examine the learned behavior of an agent trained with the lifelong intrinsic rewards only (see Fig. 2), by rolling out the learned policies in the environment. We find that the trained agent is often stuck in the first room and oscillating between two states. An example, as shown in Fig. 6, is that the agent keeps opening and closing the first door. In comparison, the agent trained with episodic intrinsic rewards only can explore all rooms without oscillating (see videos in the supplementary material). A possible reason for this failure could be that the lifelong intrinsic reward

Figure 6: The agent gets stuck, keeping opening and closing the door.

is not able to accurately reflect the novelty of states. During training, the agent may randomly sample a moving-forward action after it opens the door, but the lifelong intrinsic reward it receives from this step is lower than the previous opening-door step. Thus, as the training progresses, the agent gets stuck in the opening-closing behavior since this gives higher intrinsic rewards than moving forward to explore new rooms.

To verify this, we randomly permute the lifelong intrinsic rewards within a batch, and evaluate the resulting performance under sparse extrinsic rewards. Specifically, for a transition batch $\{(s_j, a_j, s'_j)\}_{j=1}^{B}$ and the corresponding intrinsic reward batch $\{r_j\}_{j=1}^{B}$, we randomly permute $\{r_j\}_{j=1}^{B}$, such that each transition $(s_j, a_j, s'_j)$ might be associated with the wrong intrinsic reward $r_k$ (where $k \neq j$) rather than the correct one. If this random permutation does not lead to a significant performance drop, then it indicates that the computed lifelong intrinsic reward may not accurately measure the novelty of states. From results in Fig. 5, we can see that permutation does not have much negative impact on the performance for most cases, especially when using $r^{\text{ep-visit}}$ as episodic intrinsic reward. This shows that the computed lifelong intrinsic reward may not accurately reflect the novelty of states, otherwise such permutation would lead to an obvious performance drop. We note the performance impact is not negligible in one case (*e.g.*, BeBold + ep_count in MultiRoom environments), but we believe it will not alter the message that the lifelong intrinsic reward is not critical to performance.

## 4 DISCUSSIONS

In the above experiments, we show that episodic intrinsic reward solely is able to boost exploration in procedurally-generated grid-world environments, while lifelong intrinsic reward struggles to make progress. Such intriguing observations provide new perspectives to think about intrinsic reward design as below.

**Episodic intrinsic reward in latent spaces** Our experiments use grid-world environments as testbeds, which have discrete state spaces, so it is easy to obtain visitation counts for each state. Therefore, one may wonder whether episodic intrinsic reward is able to generalize to high-dimensional continuous state spaces, where it is infeasible to calculate $r^{\text{ep-visit}}$ and $r^{\text{ep-count}}$. One possible solution is to compute the episodic intrinsic reward in a latent space (Badia et al., 2019). However, we empirically find that directly borrowing this solution from singleton environments to MiniGrid is ineffective, as latent embedding learning is much more challenging in procedurally-generated environments. For future research, we may consider designing more powerful embedding models.

**Lifelong intrinsic reward in procedurally-generated environments** Lifelong intrinsic reward like RND (Burda et al., 2019b) and ICM (Pathak et al., 2017) has shown promising performance in singleton environments (*e.g.*, Atari (Bellemare et al., 2013)), but performs poorly in procedurally-generated environments (*e.g.*, MiniGrid). This counter-intuitive phenomenon further shows that quality of the representation plays an important role in the success of curiosity-driven methods. To make lifelong intrinsic reward work again, one has to think about more generalizable representation learning for RL. One possible direction is to design proper auxiliary or pretext tasks like self-supervised learning instead of just memorizing the environment.

## 5 RELATED WORKS

**Exploration and intrinsic rewards** In RL, the agent improves its policy via trial and error. Thus a steady stream of reward signals is critical for efficient learning. Since many practical scenarios come

with sparse rewards, studying how to encourage exploration behavior in such environments attracts continuing attention. One of the most popular techniques is to provide intrinsic rewards (Schmidhuber, 2010), which can viewed as the agent's curiosity about particular states. In the absence of extrinsic rewards from the environment, the agent's behavior will be driven by the intrinsic rewards. If we expect the agent to explore novel states, the intrinsic reward can be designed to reflect how different the states are from those already visited. Such difference can be measured by the pseudo-counts of states derived from a density model (Bellemare et al., 2016; Ostrovski et al., 2017), or errors of predicting the next state (Pathak et al., 2017) or a random target (Burda et al., 2019b).

**Episodic and lifelong intrinsic reward** Prior works (Pathak et al., 2017; Burda et al., 2019a;b) typically model a lifelong intrinsic reward, which is updated throughout the whole learning process. Recent works focusing on procedurally-generated environments (Raileanu & Rocktäschel, 2020; Zhang et al., 2020) additionally use episodic intrinsic rewards to modulate the lifelong ones. However, they mainly focus on designing the lifelong intrinsic rewards. In this work we show that the episodic intrinsic reward is actually more important in encouraging exploration. Concurrent to our work, Henaff et al. (2022) also finds that the episodic intrinsic reward is essential to good performance.

Our work is also related to recent progress on Atari (Badia et al., 2019; 2020), where more attention is paid to episodic intrinsic rewards and the lifelong intrinsic reward is only considered as an optional modulator. They focus on pushing performance limits on Atari, while our work aims to compare the individual contributions of episodic and lifelong intrinsic rewards in procedurally-generated gridworld environments. One interesting connection to note is that the episodic visitation count bonus considered in our paper is essentially an MBIE-EB-style bonus (Strehl & Littman, 2008).

**Generalization and procedurally-generated environments** Recent papers (Rajeswaran et al., 2017; Zhang et al., 2018a;b;c; Machado et al., 2018; Song et al., 2020) find that deep RL is susceptible to overfitting to training environments and suffers poor generalization. Using procedurally-generated environments (Chevalier-Boisvert et al., 2018; Chevalier-Boisvert, 2018; Juliani et al., 2019; Cobbe et al., 2020), where each episode is randomly constructed but corresponds to the same task. To make progress in such environments, an agent must learn a generic policy that can generalize to novel episodes. Most prior works in exploration are also susceptible to overfitting issues, since they train and test on the same environment, such as *Montezuma's Revenge* (Bellemare et al., 2013) or *VizDoom* (Wydmuch et al., 2018). To achieve generalizable exploration, recent works (Raileanu & Rocktäschel, 2020; Campero et al., 2021; Zhang et al., 2020; Zha et al., 2021; Flet-Berliac et al., 2021) often use procedurally-generated gridworld environments to benchmark their methods.

# 6 CONCLUSIONS

Exploration is a long-standing and important problem in reinforcement learning, especially when the environments have sparse rewards. To encourage exploration in procedurally-generated environments, recent works combine lifelong and episodic intrinsic rewards. However, how these two kinds of intrinsic rewards actually contribute to exploration is seldom comprehensively investigated. To answer this question, we conducted the first systematic empirical study by making exhaustive combinations of lifelong and episodic intrinsic rewards in environments with and without extrinsic reward signals. The results are counter-intuitive and intriguing: episodic intrinsic reward alone is able to boost exploration and set new state-of-the-art. Our further investigation shows that the poor performance of lifelong intrinsic reward roots in its ineffectiveness in distinguishing whether a state is novel or not. We believe our findings are inspiring for future work to better design intrinsic reward functions. We would like to note that our discovery focuses more on hard exploration tasks in MiniGrid, and whether it applies to general procedurally generated environments requires further investigation.

# 7 REPRODUCIBILITY STATEMENT

We describe the training details (*e.g.*, environment configurations, network architectures, hyper-parameters) in Sec. 3.1 and the Appendix. To reproduce the results, we also include the source code in the supplementary material. The computation cost is detailed in Appx A.3.

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

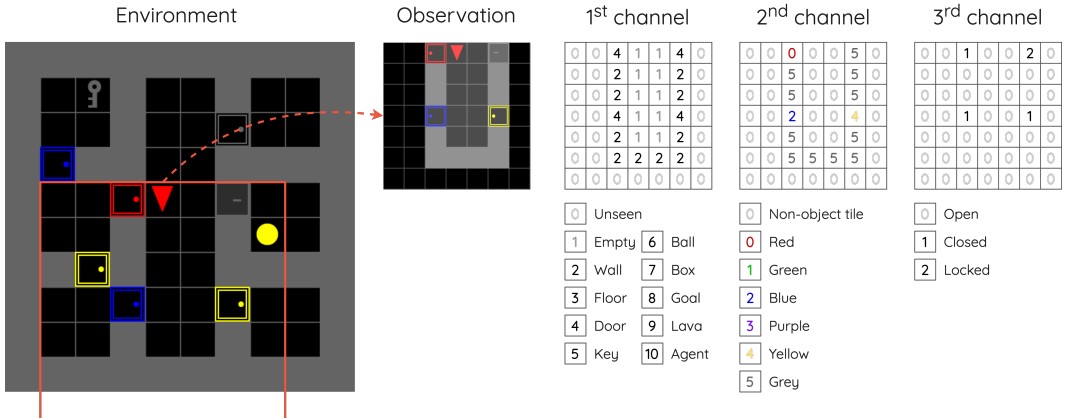

Figure 7: Observation encoding in MiniGrid environment. See text for details. Best viewed in color.

## A    APPENDIX

### A.1    MINIGRID ENVIRONMENT

MiniGrid (Chevalier-Boisvert et al., 2018) benchmark consists a suite of procedurally-generated gridworld environments. The gridworld is built with $N \times N$ tiles, where $N$ varies across tasks. Each tile contains one or no object. The possible object types include wall, floor, lava, door, key, ball, box and goal.

**Observation**    By default, the observation is an egocentric view of the environment, as shown in Fig. 7. The agent can not see the world behind walls or closed doors. Specifically, the observation is compactly encoded as a $7 \times 7 \times 3$ array. The first channel encodes the object type, *e.g.*, wall, door. The second channel encodes the object color, *e.g.*, red, green. The third channel encodes the state, *e.g.*, door open, door closed.

**Action**    In MiniGrid, there are 7 actions available for the agent: *turn left*, *turn right*, *move forward*, *pick up an object*, *drop an object*, *toggle* and *done*. The agent can change the direction it is currently facing by turning left or right. Taking the *move forward* action will move the agent one tile forward along its current direction. If moving forward causes collision (*e.g.*, running into walls or closed doors), the agent will remain the previous location. The agent can use the *toggle* action to open a closed door in front of it (or a locked one if the agent has the corresponding key). The *toggle* action can also be used to open a box. The *done* action is not necessary for the environments considered in this paper, but we still keep it in the action space following previous works.

**Task**    As briefly introduced in Sec. 3.1, we consider 3 tasks in MiniGrid: `KeyCorridor`, `MultiRoom`, `ObstructedMaze`. Here we provide more details about these 3 tasks.

- `KeyCorridor` (Fig. 8) The environment consists of a corridor and several rooms connecting to the corridor. The agent starts in the corridor and aims to pick up a goal object placed in one of the rooms. The door to this room (*i.e.*, the room with the goal) is locked. To open the locked door, the agent has to find the key that is placed in one of the rooms except the locked one. The configuration suffix is specified as `SxRy`, where `x` denotes the room size and `y` denotes the number of rows. Each row has 2 rooms, placed on each side of the corridor.
- `MultiRoom` (Fig. 9) The environment consists of a sequence of rooms connected by closed doors. The agent starts in the first room and aims to reach the goal position in the last room. The configuration suffix is specified as `NxSy`, where `x` denotes the number of rooms and `y` denotes the maximum room size.
- `ObstructedMaze` (Fig. 10) The environment consists of a $3 \times 3$ grid of rooms. In these 9 rooms, 1 is the center room, 4 are corner rooms and 4 are non-corner rooms. The doors connecting the center room and non-corner rooms are closed. The doors connecting corner rooms and

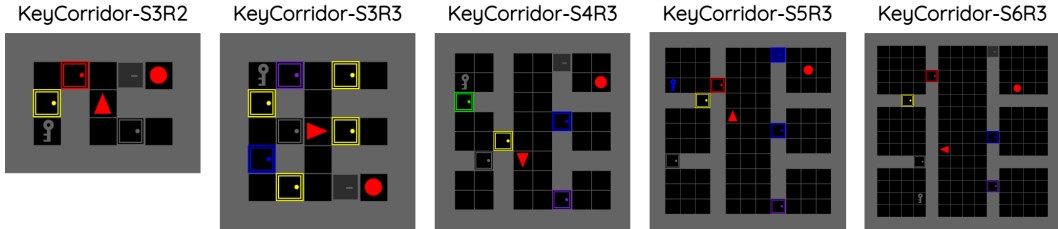

Figure 8: `KeyCorridor` environments.

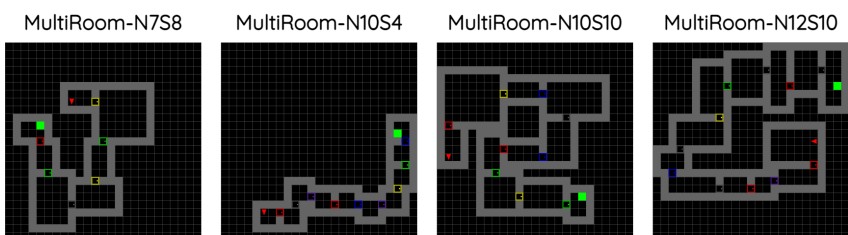

Figure 9: `MultiRoom` environments.



Figure 10: `ObstructedMaze` environments.

non-corner rooms are locked. The keys are placed in non-corner rooms. Depending on the configuration, the agent starts in the center room or non-corner rooms, and aims to pick up an goal object hidden in one of the corner rooms. The configuration suffix is more complex, so we only introduce commonly used ones and refer interested readers to the official GitHub repository[1].

- `2Dlh` The goal object is always hidden in the top right room. The keys to the corner rooms are hidden in boxes. The agent always starts in the middle right room.
- `2Dlhb` Compared to `2Dlh`, the difference is that locked doors are blocked by obstructions. The agent has to move these obstructions before unlocking the doors with keys.
- `1Q` Compared to `2Dlhb`, the difference is that the agent starts in the center room instead of the middle right one.
- `2Q` Compared to `1Q`, the difference is that the goal object is hidden in either the top right room or the bottom right room.
- `Full` Compared to `1Q`, the difference is that the goal object is hidden in one of the 4 corner rooms.

**Episode length**  In these tasks, there is a limit on the maximum number of steps allowed in each episode, denoted as $t_{\max}$. It varies from task to task, as summarized in Fig. 11.

**Reward**  When the agent achieves the goal, it receives an extrinsic reward $r_t^e$. This reward is computed based on the number of steps taken $t$ and the maximum number of steps allowed $t_{\max}$:

---

[1]https://github.com/maximecb/gym-minigrid

| Task | MultiRoom | | | | | | KeyCorridor | | | | | ObstructedMaze | | | | |
|---|---|---|---|---|---|---|---|---|---|---|---|---|---|---|---|---|
| Suffix | N6S10 | N7S4 | N7S8 | N10S4 | N10S10 | N12S10 | S3R2 | S3R3 | S4R3 | S5R3 | S6R3 | 2Dlh | 2Dlhb | 1Q | 2Q | Full |
| $t_{\max}$ | 120 | 140 | 140 | 200 | 200 | 240 | 270 | 270 | 480 | 750 | 1080 | 576 | 576 | 720 | 1584 | 3600 |
| Frames | 2e7 | 2e7 | 2e7 | 2e7 | 2e7 | 2e7 | 2e7 | 2e7 | 4e7 | 4e7 | 6e7 | 4e7 | 6e7 | 6e7 | 9e7 | 1e8 |
| RIDE | ✘ | ✔ | ✔ | ✔ | ✔ | ✔ | ✘ | ✔ | ✘ | ✘ | ✘ | ✔ | ✘ | ✘ | ✘ | ✘ |
| BeBold | ✔ | ✘ | ✔ | ✘ | ✘ | ✔ | ✘ | ✔ | ✔ | ✔ | ✔ | ✔ | ✔ | ✔ | ✔ | ✔ |
| RAPID | ✘ | ✔ | ✔ | ✔ | ✔ | ✔ | ✔ | ✔ | ✔ | ✘ | ✘ | ✘ | ✘ | ✘ | ✘ | ✘ |
| AGAC | ✘ | ✘ | ✘ | ✘ | ✔ | ✘ | ✘ | ✘ | ✔ | ✔ | ✘ | ✘ | ✔ | ✔ | ✔ | ✘ |

Figure 11: Different MiniGrid task configurations used in previous works.

$r_t^e = 1 - 0.9t/t_{\max}$. The agent is encouraged to reach the goal as fast as possible. Only the optimal path gives the highest reward.

## A.2 NETWORK ARCHITECTURES

For the policy and value network in PPO, we use a shared feature extractor and two separate heads (one policy head and one value head), as shown in Fig. 12 (a). The feature extractor uses a convolutional neural network, similar to the ones used in prior works (Raileanu & Rocktäschel, 2020; Zhang et al., 2020; Flet-Berliac et al., 2021). For $r^{\text{RND}}$ and $r^{\text{BeBold}}$, the target and predictor networks use the architecture in Figure 12 (b). For $r^{\text{ICM}}$ and $r^{\text{RIDE}}$, the network architecture of the state embedding function $f_{\text{emb}}$ is shown in Figure 12 (b). The network architectures of the forward and inverse dynamic model are shown in Figure 12 (c) and Figure 12 (d) respectively.

## A.3 COMPUTATION COST

As mentioned in Sec. 3.1, our experiments can run on a single machine with 8 CPUs and a Titan X GPU. Specifically, one run of an experiment (*e.g.*, training with $r^{\text{ICM}} + r^{\text{ep-count}}$) takes less than 600M GPU memory on a Titan X GPU. So, we can concurrently run several experiments on a single GPU. On average, training for 1e7 frames takes about 1.5 hours. The total wall-clock training time for all conducted experiments is roughly 4000 GPU hours.

In comparison, the IMPALA used in the released code of RIDE (Raileanu & Rocktäschel, 2020) is typically 10-20x slower than PPO for our hardware. The low efficiency is mainly because it uses CPU to do the forward pass of the policy during experience collection and it is non-trivial for us to migrate it to GPU. This also means that we need a large number of CPUs (typically 100+) to collect samples in parallel to reach the full speed. Such computation requirement (high CPU but low GPU demand) does not fit our computation resources (where each workstation often has less than 32 CPUs). Therefore, we choose PPO as the base algorithm to fully utilize our resources.

## A.4 TRAINING DETAILS

We provide more training details for the comparison with other state-of-the-art methods in Sec. 3.3.

For RIDE (Raileanu & Rocktäschel, 2020), we obtained results on `MultiRoom-N7S8`, `MultiRoom-N10S10` from the authors, and rerun their released code[2] for other environments. For `KeyCorridor-S4R3` and `KeyCorridor-S6R3`, we use the same hyperparameters as `KeyCorridor-S3R3` provided in their paper. For `ObstructedMaze-1Q` and `ObstructedMaze-2Q`, we use the same hyperparameters as `ObstructedMaze-2Dlh`.

For AGAC (Flet-Berliac et al., 2021), we obtained results on `MultiRoom-N10S10`, `ObstructedMaze-1Q` and `ObstructedMaze-2Q` from the authors, and rerun their released code[3] for other environments. For hyperparameters, we use their default configurations.

---

[2] https://github.com/facebookresearch/impact-driven-exploration
[3] https://github.com/yfletberliac/adversarially-guided-actor-critic

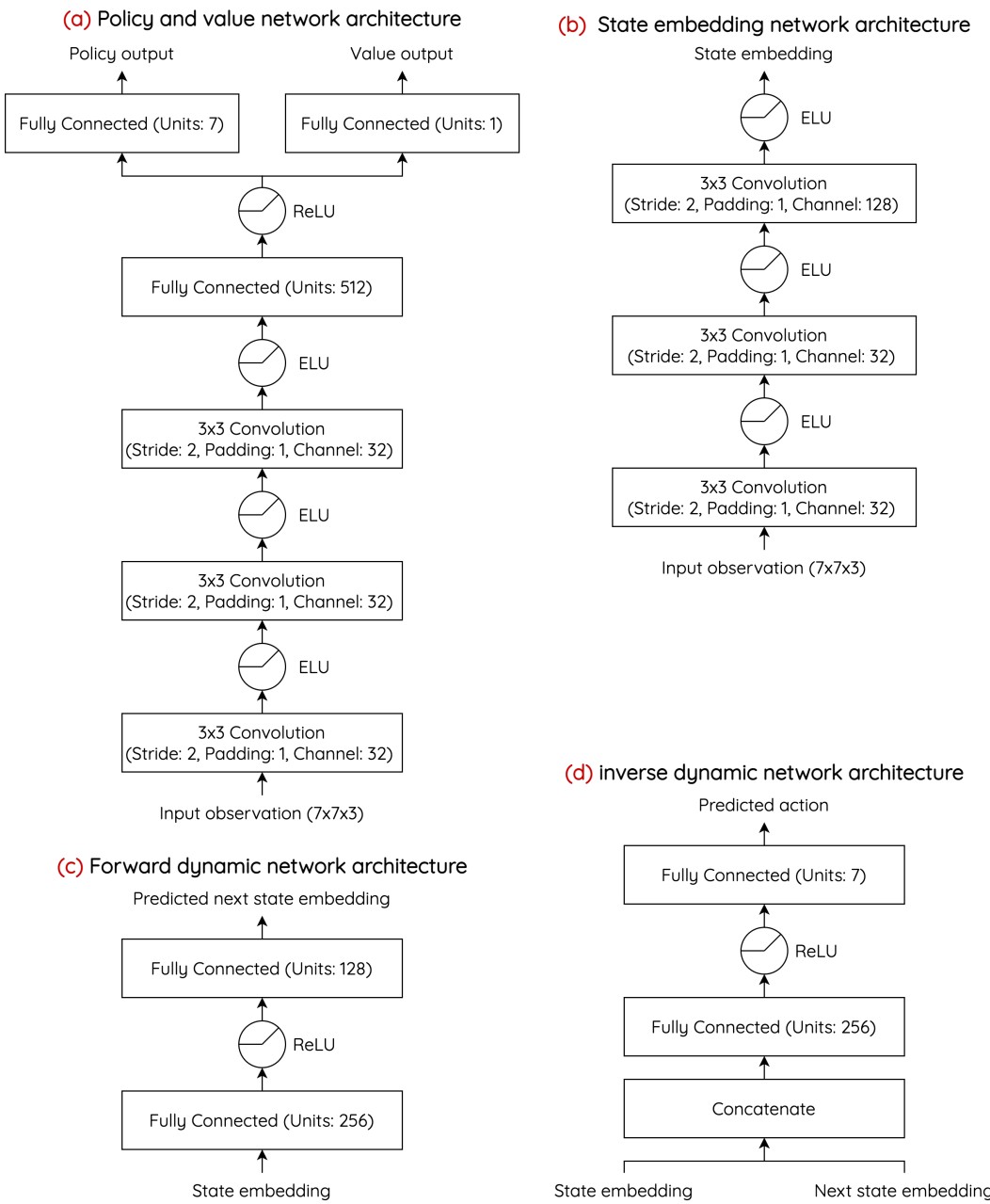

Figure 12: Network architectures.

For RAPID (Flet-Berliac et al., 2021), we obtained results on `MultiRoom-N7S8` and `MultiRoom-N10S10` from the authors, and rerun their released code[4] for other environments. For hyperparameters, we use their default configurations.

For BeBold (Zhang et al., 2020), we obtained the results on `KeyCorridor-S4R3`, `KeyCorridor-S6R3`, `MultiRoom-N7S8`, `ObstructedMaze-1Q` and `ObstructedMaze-2Q` from the authors, and rerun their released code[5] for `MultiRoom-N10S10`. For hyperparameters, we use their default configurations.

---

[4]https://github.com/daochenzha/rapid
[5]https://github.com/tianjunz/NovelD

## A.5 HYPERPARAMETERS

For training the policy, we use the hyperparameters listed in Tab. 1. Generalized Advantage Estimation (GAE) (Schulman et al., 2016) is used to estimate the advantage. As mentioned in Sec. 3.1, we search the intrinsic reward coefficient $\beta$ for each experiment. We start with a search range {5e-2, 1e-2, 5e-3, 1e-3}, and expand it until finding a locally best value or reaching 5e1 or 1e-5. The best searched values for $\beta$ are summarized in Fig. 13, 14, 15, 16, 17. Here we showcase in Fig. 18 that choosing an appropriate value for $\beta$ is critical for the performance. Other results in the search (*i.e.*, using different $\beta$) will be made publicly available.

Table 1: Hyperparameters.

| | |
|---|---|
| Number of parallel environments | 128 |
| Number of timesteps per rollout | 128 |
| PPO clip range | 0.2 |
| Discount factor $\gamma$ | .99 |
| GAE $\lambda$ | .95 |
| Number of epochs | 4 |
| Number of minibatches per epoch | 8 |
| Entropy bonus coefficient | 0.01 |
| Value loss coefficient | 0.5 |
| Advantage normalization | Yes |
| Gradient clipping ($\ell_2$ norm) | 0.5 |
| Learning rate | $5 \times 10^{-4}$ |
| Total timesteps | See Fig. 11 |
| Forward dynamic loss coefficient | 10 |
| Inverse dynamic loss coefficient | 0.1 |
| RND loss coefficient | 0.1 |

| Experiments \ Environments | KeyCorridor | | MultiRoom | | ObstructedMaze | |
|---|---|---|---|---|---|---|
| | S4R3 | S6R3 | N7S8 | N10S10 | 1Q | 2Q |
| $r^{\text{ICM}}$ only | 1e-1 | 1e-1 | 1e-1 | 1e-1 | 1e-1 | 1e-1 |
| $r^{\text{ICM}} + r^{\text{ep\_count}}$ | 5e-1 | 5e-1 | 1e-1 | 1e-1 | 1e-1 | 1e-1 |
| $r^{\text{ICM}} + r^{\text{ep\_visit}}$ | 1e-1 | 5e-1 | 1e-1 | 1e-1 | 1e-1 | 1e-1 |
| $r^{\text{ICM}}$ (permuted) $+ r^{\text{ep\_count}}$ | 5e-1 | 1 | 5e-1 | 5e-1 | 1e-1 | 1e-1 |
| $r^{\text{ICM}}$ (permuted) $+ r^{\text{ep\_visit}}$ | 5e-1 | 5e-1 | 5e-1 | 5e-1 | 5e-2 | 5e-2 |

Figure 13: Best searched $\beta$ values for experiments with $r^{\text{ICM}}$.

| Experiments \ Environments | KeyCorridor | | MultiRoom | | ObstructedMaze | |
|---|---|---|---|---|---|---|
| | S4R3 | S6R3 | N7S8 | N10S10 | 1Q | 2Q |
| $r^{\text{RIDE}}$ only | 1e-1 | 1e-1 | 1e-1 | 1e-1 | 1e-1 | 1e-1 |
| $r^{\text{RIDE}} + r^{\text{ep\_count}}$ | 5e-1 | 5e-1 | 5e-2 | 5e-1 | 5e-2 | 5e-1 |
| $r^{\text{RIDE}} + r^{\text{ep\_visit}}$ | 5e-1 | 5e-1 | 1e-1 | 1e-1 | 5e-2 | 5e-2 |
| $r^{\text{RIDE}}$ (permuted) $+ r^{\text{ep\_count}}$ | 1 | 1 | 5e-1 | 5e-1 | 1e-1 | 1e-1 |
| $r^{\text{RIDE}}$ (permuted) $+ r^{\text{ep\_visit}}$ | 5e-1 | 5e-1 | 5e-1 | 5e-1 | 5e-2 | 1e-1 |

Figure 14: Best searched $\beta$ values for experiments with $r^{\text{RIDE}}$.

| Environments / Experiments | KeyCorridor | | MultiRoom | | ObstructedMaze | |
|---|---|---|---|---|---|---|
| | S4R3 | S6R3 | N7S8 | N10S10 | 1Q | 2Q |
| $r^{\mathrm{RND}}$ only | 1e-3 | 1e-3 | 1e-3 | 1e-3 | 1e-3 | 1e-3 |
| $r^{\mathrm{RND}} + r^{\mathrm{ep\_count}}$ | 5e-4 | 1e-3 | 5e-4 | 5e-4 | 1e-5 | 5e-5 |
| $r^{\mathrm{RND}} + r^{\mathrm{ep\_visit}}$ | 5e-3 | 5e-3 | 5e-3 | 5e-3 | 1e-5 | 5e-5 |
| $r^{\mathrm{RND}}$ (permuted) $+ r^{\mathrm{ep\_count}}$ | 1e-3 | 1e-3 | 1e-3 | 1e-3 | 1e-4 | 1e-4 |
| $r^{\mathrm{RND}}$ (permuted) $+ r^{\mathrm{ep\_visit}}$ | 5e-3 | 5e-3 | 5e-3 | 5e-3 | 1e-4 | 1e-4 |

Figure 15: Best searched $\beta$ values for experiments with $r^{\mathrm{RND}}$.

| Environments / Experiments | KeyCorridor | | MultiRoom | | ObstructedMaze | |
|---|---|---|---|---|---|---|
| | S4R3 | S6R3 | N7S8 | N10S10 | 1Q | 2Q |
| $r^{\mathrm{BeBold}}$ only | 5e-3 | 5e-3 | 5e-3 | 5e-3 | 5e-3 | 5e-3 |
| $r^{\mathrm{BeBold}} + r^{\mathrm{ep\_count}}$ | 1e-2 | 1e-2 | 5e-2 | 1e-2 | 1e-3 | 5e-3 |
| $r^{\mathrm{BeBold}} + r^{\mathrm{ep\_visit}}$ | 1e-1 | 5e-2 | 5e-2 | 5e-2 | 5e-3 | 5e-3 |
| $r^{\mathrm{BeBold}}$ (permuted) $+ r^{\mathrm{ep\_count}}$ | 1e-1 | 1e-1 | 1e-2 | 5e-3 | 5e-3 | 5e-3 |
| $r^{\mathrm{BeBold}}$ (permuted) $+ r^{\mathrm{ep\_visit}}$ | 1e-1 | 1e-1 | 1e-1 | 1e-1 | 5e-3 | 1e-2 |

Figure 16: Best searched $\beta$ values for experiments with $r^{\mathrm{BeBold}}$.

| Environments / Experiments | KeyCorridor | | | | MultiRoom | | | | ObstructedMaze | | |
|---|---|---|---|---|---|---|---|---|---|---|---|
| | S3R3 | S4R3 | S5R3 | S6R3 | N7S8 | N10S4 | N10S10 | N12S10 | 2Dlh | 1Q | 2Q |
| $r^{\mathrm{ep\_count}}$ only | 1e-3 | 1e-3 | 1e-3 | 1e-3 | 1e-3 | 1e-3 | 1e-3 | 1e-3 | 5e-5 | 5e-4 | 5e-4 |
| $r^{\mathrm{ep\_visit}}$ only | 5e-3 | 5e-3 | 5e-3 | 5e-3 | 5e-3 | 5e-3 | 5e-3 | 1e-3 | 5e-5 | 5e-5 | 5e-5 |

Figure 17: Best searched $\beta$ values for experiments with episodic intrinsic rewards only.

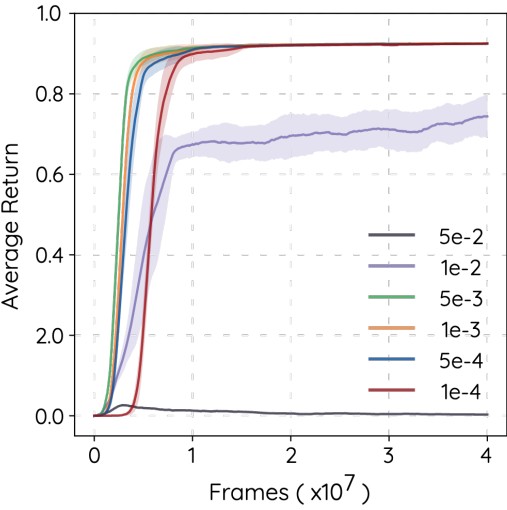

Figure 18: Performance of training with $r^{\mathrm{ep\text{-}visit}}$ only on `KeyCorridorS4R3`, using different $\beta$.

### A.6 ADDITIONAL RESULTS

#### A.6.1 FINAL TESTING PERFORMANCE OF TRAINED AGENTS

For completeness, we test the trained agent for 100 episodes and show the results in Tab. 2.

Table 2: Final testing performance of trained agents.

| | KeyCorridor | | MultiRoom | | ObstructedMaze | |
| | S4R3 | S6R3 | N7S8 | N10S10 | 1Q | 2Q |
|---|---|---|---|---|---|---|
| $r^{\text{ep\_count}}$ only | 0.93±0.026 | 0.95±0.034 | 0.68±0.006 | 0.69±0.005 | 0.85±0.241 | 0.71±0.388 |
| $r^{\text{ep\_visit}}$ only | 0.93±0.020 | 0.95±0.026 | 0.68±0.001 | 0.69±0.009 | 0.83±0.280 | 0.66±0.391 |
| $r^{\text{ICM}}$ only | 0.0±0.0 | 0.0±0.0 | 0.0±0.0 | 0.0±0.0 | 0.0±0.0 | 0.0±0.0 |
| $r^{\text{ICM}} + r^{\text{ep\_count}}$ | 0.93±0.020 | 0.94±0.046 | 0.67±0.008 | 0.69±0.003 | 0.84±0.248 | 0.73±0.351 |
| $r^{\text{ICM}} + r^{\text{ep\_visit}}$ | 0.93±0.025 | 0.95±0.027 | 0.63±0.054 | 0.69±0.002 | 0.85±0.253 | 0.64±0.401 |
| $r^{\text{RIDE}}$ only | 0.0±0.0 | 0.0±0.0 | 0.0±0.0 | 0.0±0.0 | 0.0±0.0 | 0.0±0.0 |
| $r^{\text{RIDE}} + r^{\text{ep\_count}}$ | 0.93±0.023 | 0.92±0.133 | 0.66±0.023 | 0.69±0.006 | 0.86±0.241 | 0.61±0.417 |
| $r^{\text{RIDE}} + r^{\text{ep\_visit}}$ | 0.93±0.022 | 0.95±0.023 | 0.43±0.022 | 0.69±0.005 | 0.79±0.322 | 0.0±0.0 |
| $r^{\text{RND}}$ only | 0.0±0.0 | 0.0±0.0 | 0.0±0.0 | 0.0±0.0 | 0.0±0.0 | 0.0±0.0 |
| $r^{\text{RND}} + r^{\text{ep\_count}}$ | 0.93±0.032 | 0.93±0.099 | 0.56±0.100 | 0.69±0.005 | 0.83±0.227 | 0.61±0.408 |
| $r^{\text{RND}} + r^{\text{ep\_visit}}$ | 0.92±0.028 | 0.93±0.097 | 0.67±0.008 | 0.68±0.001 | 0.54±0.420 | 0.72±0.362 |
| $r^{\text{BeBold}}$ only | 0.0±0.0 | 0.0±0.0 | 0.0±0.0 | 0.0±0.0 | 0.0±0.0 | 0.0±0.0 |
| $r^{\text{BeBold}} + r^{\text{ep\_count}}$ | 0.92±0.068 | 0.93±0.047 | 0.68±0.009 | 0.69±0.006 | 0.0±0.0 | 0.0±0.0 |
| $r^{\text{BeBold}} + r^{\text{ep\_visit}}$ | 0.93±0.024 | 0.94±0.033 | 0.66±0.022 | 0.69±0.015 | 0.73±0.309 | 0.60±0.395 |

#### A.6.2 COMPARISON TO STATE-OF-THE-ART METHODS ON OTHER ENVIRONMENTS IN MINIGRID

In addition to the comparison in Sec. 3.3, we provide results on following environments: `KeyCorridorS3R3`, `KeyCorridorS5R3`, `MultiRoomN10S4`, `MultiRoomN12S10`, `ObstructedMaze2Dlh`. The results in Fig. 19 further demonstrates that $r^{\text{ep-count}}$ or $r^{\text{ep-visit}}$ can match or surpass other state-of-the-art methods.

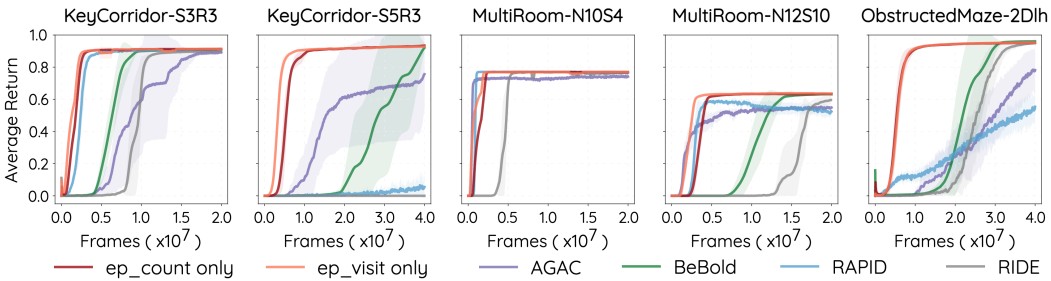

Figure 19: Comparison of only using episodic intrinsic rewards and state-of-the-art methods on additional environments. Best viewed in color.

#### A.6.3 COMPARISON OF LIFELONG COUNT-BASED BONUS AND EPISODIC VISITATION BONUS

During the discussion period, we conduct experiments of only using lifelong count-based intrinsic rewards $\frac{1}{\sqrt{N_{life}(s_{t+1})}}$ (instead of episodic ones). The results are shown in Fig. 20. We can see the performance is close to that of only using $r^{\text{ep-visit}}$. One possible reason is that lifelong counts behave quite similarly to episodic counts since every episode is new in procedurally-generated environments. It is also possible that the performance difference between episodic and lifelong intrinsic rewards on MiniGrid actually results from "count-based intrinsic rewards" v.s. "prediction-based intrinsic rewards".

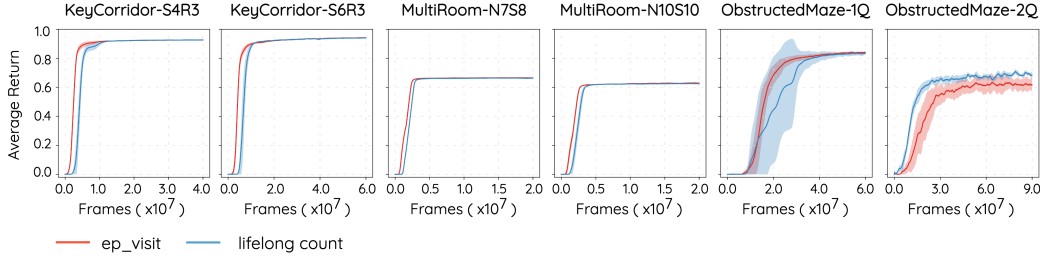

Figure 20: Comparison to lifelong count-based bonus $\dfrac{1}{\sqrt{N_{life}(s_{t+1})}}$ on 6 hard exploration environments in MiniGrid, averaged from 5 runs. Best viewed in color.

### A.6.4  MAIN RESULTS AVERAGED FROM 10 RUNS

During the discussion period, we conduct the main experiments (Fig. 2) with additional runs and plot the results in Fig. 21.

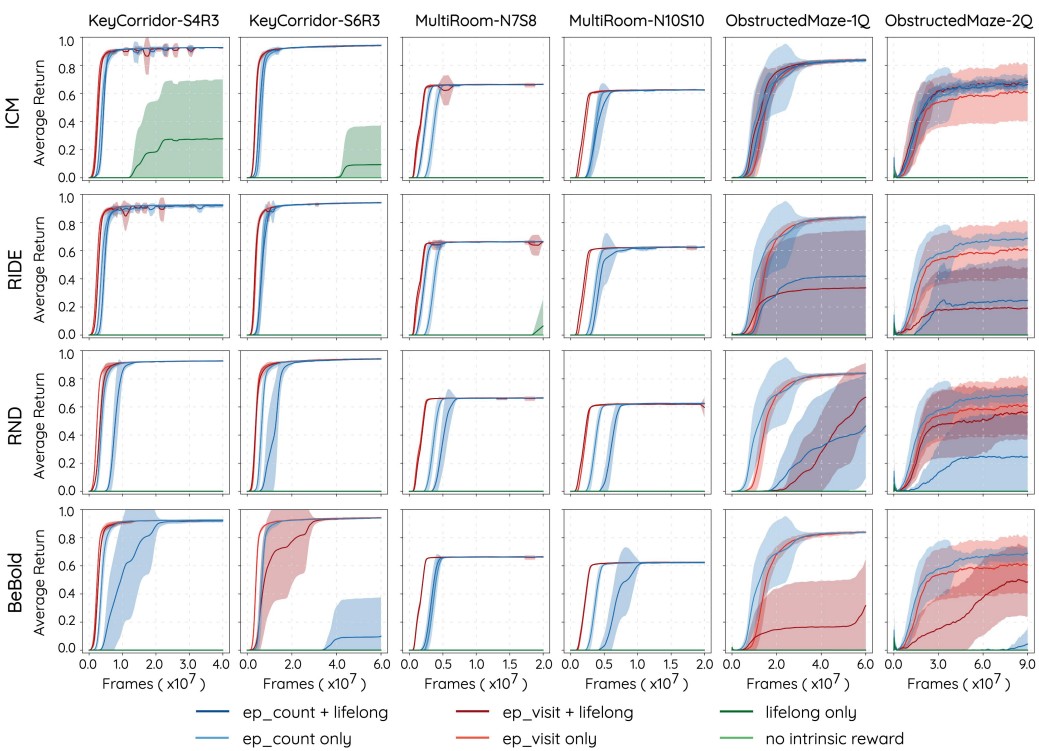

Figure 21: Performance of different combinations of lifelong and episodic intrinsic rewards on 6 hard exploration environments in MiniGrid, averaged from 10 runs. Note that simply using episodic intrinsic rewards can achieve top performance among all combinations. Best viewed in color.

