# OpenReview forum: "Revisiting Intrinsic Reward for Exploration in Procedurally Generated Environments"
_ICLR.cc/2023/Conference — ICLR 2023 poster_

### Official Review · Reviewer_bLVL · 2022-10-13

**Confidence:** 4
**Correctness:** 3
**Technical Novelty And Significance:** 2
**Empirical Novelty And Significance:** 1
**Recommendation:** 3

**Clarity, Quality, Novelty And Reproducibility:**

The arguments of the paper are clearly stated with various experiments. The originality of the paper is weak since the paper just extracted episodic curiosity term from RIDE and BeBold and apply it to other curiosity models or alone.



**Strength And Weaknesses:**

**[Strength]**

1. The paper is well written and easy to understand, especially the difference between each existing method.

2. Moreover, the authors presented that a simple count-based method in each episode is already enough for training MiniGrid.


**[Weakness]**

1. The authors omitted a highly relevant paper [Ref 1]. The author should compare with [Ref 1] in terms of performance and conceptual difference. Moreover, It could be the answer to the first question in the Discussion section.

2. As already mentioned in the Discussion section, the authors only used MiniGrid which is a discrete and almost static environment. It seems that the reason why episodic curiosity was effective is that the action and the position of agents are discrete and the number of possible positions is relatively small compared to other environments such as ProcGen (Cobbe 2020). Consequently, explicitly using the state visitation information is effective and not surprising. In other words, from the current version of the paper, it is difficult to the convenience that episodic curiosity is really effective in other procedurally generated environments.


[Ref. 1] Savinov, Nikolay, et al. "Episodic curiosity through reachability." ICLR. 2019.


**[Minor]**

On page 7, the author mentioned that the official implementation of BeBold is not open-sourced. However, the first author of that paper released the code in November 2021 (tianjunz/NovelD).

"Leveraging procedural generation to benchmark reinforcement learning" has two BibTeX entries (Cobbe 2019, Cobbe 2020). Please merge them together.

In README.md, "NeurIPS 2021" -> "ICLR 2023"

**Summary Of The Paper:**

The paper analyzes various intrinsic reward generation algorithms in the MiniGrid benchmark and presents that episodic curiosity is helpful for training while lifelong curiosity is not.

The authors compare four different curiosity algorithms (ICM, RIDE, RND, BeBold) with and without two episodic curiosity (episodic visitation count from RIDE and episodic first visits from BeBold). The experimental results show that episodic curiosity-only models generally have the best performance while models with the lifelong curiosity-only have the worst performance and models using both curiosities have similar or worse performance than the lifelong curiosity-only model.

The authors supposed that this might be because lifelong curiosity cannot capture the correct novelty of the state presenting that randomly shuffled reward within each batch did not hugely deter the performance.


**Summary Of The Review:**

Although analyzing the effectiveness of lifelong curiosity and episodic curiosity is interesting, as mentioned in the weakness section, not compared with [Ref. 1] and the method is only applicable to discrete environments, not the general procedurally-generated environments, I gave Reject as the result of the initial review.

---

> ### Author Response · Authors · 2022-11-10
> **Authors' response**
>
> Thank the reviewer for the constructive feedback. We are glad to hear that our paper is well-written and easy to understand. Below we respond to the comments. We also upload a revised version of the paper that incorporates the suggested changes.
>
> > `> Comment 1 (C1):` The authors omitted a highly relevant paper [Ref 1]. The author should compare with [Ref 1] in terms of performance and conceptual difference. Moreover, I could be the answer to the first question in the Discussion section.
>
> `> Response 1 (R1):` Thank the reviewer for bringing up this relevant paper. We are currently adapting it to our code base and running experiments. We will update the results soon and add discussions about the comparison with it.
>
> > `> C2:` As already mentioned in the Discussion section, the authors only used MiniGrid which is a discrete and almost static environment. It seems that the reason why episodic curiosity was effective is that the action and the position of agents are discrete and the number of possible positions is relatively small compared to other environments such as ProcGen (Cobbe 2020). Consequently, explicitly using the state visitation information is effective and not surprising. In other words, from the current version of the paper, it is difficult to the convenience that episodic curiosity is really effective in other procedurally generated environments.
>
> `> R2:` We agree that more experiments and analysis on complex environments such as Procgen would further strengthen our paper. Our paper is more like a comprehensive empirical study on MiniGrid, which is extensively used in prior works (as the main benchmark) to study exploration in procedurally-generated environments. However, how episodic and lifelong curiosity contributes to the exploration performance on this important benchmark has not received much attention. Thus, we believe it is worthwhile sharing our observations with the community. We have revised the conclusion section to make the limitations clear, and will revise other statements that might sound like overclaims.
>
> > `> C3:` On page 7, the author mentioned that the official implementation of BeBold is not open-sourced. However, the first author of that paper released the code in November 2021 (tianjunz/NovelD).
>
> `> R3:` Thank the review for the suggestion. We are currently rerunning experiments with the NovelD code and will update the results soon.
>
> > `> C4:` Bibliography issues and typos
>
> `> R4:` Thank the reviewer for pointing them out. We have fixed them in the revised version.

---

> > ### Comment · Reviewer_bLVL · 2022-11-12
> > **Still concern about generalizability to non-discrete procedurally generated environments**
> >
> > I appreciate the authors dealing with my concerns and I will look forward to the results of the experiments mentioned by the authors.
> >
> > However, like other reviewers (3Vou, 38Yz) mentioned, I have still concerned about whether this method can be generally applicable to other procedurally-generated environments (e.g., ProcGen) or other standard benchmarks (e.g. Atari). It seems that the proposed method heavily exploits the inductive bias of the environment. I agree with the point that MiniGrid is one of the most standard benchmarks in curiosity-driven reinforcement learning. Although episodic and lifelong curiosity has not received much attention from the communities, if it requires too strict conditions (i.e., the environment should be a discrete state and action space like MiniGrid), the contribution to bringing this direction is dramatically weakened. Hence, I highly recommend authors test on ProcGen. According to the ICLR guideline, the authors can still make a comment during Discussion Stage 2 which will end Dec 12. I hope this is enough time for the authors to prepare the experiments.
> >
> > Additionally, it would be better if the authors can highlight the changes with different colors in the manuscripts so that reviewers can easily detect the changes.

---

> > > ### Author Response · Authors · 2022-11-13
> > > **Thanks for the feedback**
> > >
> > > We thank the reviewer for their feedback. We have uploaded a new version with changes highlighted.
> > >
> > > We would like to clarify that our work does not aim to propose a new "sota" method that applies to all procedurally-generated environments. Rather, our goal is to share the message that "episodic curiosity is actually what works in prior methods". The reason we use MiniGrid is simple: it is used in prior works.
> > >
> > > Regarding Procgen, it is not clear how to conduct the experiments properly. As far as we know, no existing exploration method (i.e., methods that combine lifelong and episodic curiosity) has been shown to work well in Procgen. So if we want to disentangle the lifelong and episodic curiosities, the first step is to make one method (say BeBold) work in Procgen. This is not a trivial thing to achieve. We need to design a new episodic curiosity module to replace the state count in BeBold, otherwise almost every state will be considered unique and novel. As the reviewer suggests, one idea might be learning a reachability network [1]. We are now trying it but the results so far suggest that it does not even work for MiniGrid. We will try different hyper-parameters and update all results later. In addition, we believe designing a new episodic curiosity is beyond the scope of our paper. We can try something specific (e.g., comparison with [1]) during the discussion period if the reviewer requests, but we are afraid that we do not have enough time and resources for a comprehensive study of a new episodic curiosity.
> > >
> > >
> > > [1] Savinov, Nikolay, et al. "Episodic curiosity through reachability." ICLR. 2019.

---

> > > > ### Comment · Reviewer_bLVL · 2022-11-16
> > > > **Alternative Suggestions**
> > > >
> > > > I appreciate the authors for their response and using different color for the revision
> > > >
> > > > I have acknowledged that the goal of the paper is to achieve state-of-the-art performance but deal with episodic curiosity. That is why I am worried more about the generalizability of proposed episodic curiosity to other environments.
> > > >
> > > > I understand that ProcGen is a very difficult task and other curiosity-driven reinforcement learning methods do not present the experiment. I would like to clarify my concern:
> > > >
> > > > MiniGrid has much fewer states compared to other environments such as Atari, Mujoco, and so on since it is a discrete space (grid world). In general, this was not a problem to test curiosity-driven reinforcement learning since it has sparse rewards which makes it difficult to solve the environment in some hard rooms. However, the proposed episodic count method employs the number of visits of each state in each episode. Combined with the characteristics of the environment, it can be a good solution for the environment but if the environment is stochastic and continuous, I am worried whether the episodic curiosity can work.
> > > >
> > > > Hence, if it is difficult to run on ProcGen, I have two suggestions and I hope to see both:
> > > >
> > > > * One suggestion is to test on other stochastic or continuous environments. Although Atari is not a continuous environment, it is acceptable considering the previous literature and its huge state space. They do not need to be a procedurally-generated environment unless it can resolve my worries. Even, simple environments made by authors are also appreciated.
> > > >
> > > > * Another suggestion is to compare with lifelong count-based exploration methods such as (N_lifelong)^-0.5 where N_lifelong can be actual count. Currently, the authors only compare with the prediction-based curiosity method but it is good to add a lifelong count-based method.

---

> > > > > ### Author Response · Authors · 2022-11-18
> > > > > **Thanks for the clarification and suggestions**
> > > > >
> > > > > We thank the reviewer for clarifying the concerns and offering suggestions.
> > > > >
> > > > > We will run the two suggested experiments and try our best to update the results as soon as possible. Regarding comparison to lifelong count-based exploration, the results are included in Fig. 22 in the revised version. Regarding the stochastic environments, we would like to know if the reviewer is okay with a stochastic version of MiniGrid, i.e., replacing the deterministic transitions with stochastic ones, since this requires minimal modification of our code.
> > > > >
> > > > > In addition, we would like to point out that we **do not propose** the episodic count method. Those episodic bonus terms (ep_visit and ep_count) are already used in prior works (RIDE, BeBold/NovelD, RAPID, AGAC), and what we did is conducting a comprehensive study to reveal the contributions of episodic bonuses. We believe the reviewer's comment happens to confirm that our work is worth sharing with the community. As the reviewer mentioned, given the characteristics of MiniGrid, using episodic counts can be a good solution. We find that this is actually the key factor for good performance on MiniGrid. Prior works rely on such "good solution" but mainly attribute the improvement to their proposed lifelong curiosity, which might mislead follow-up works. This is the message we would like to convey in our work.

---

> > > > > > ### Comment · Reviewer_bLVL · 2022-11-18
> > > > > > **Thanks for clarification**
> > > > > >
> > > > > > I appreciate the authors for their responses in the long discussion and for adding N_lifelong results.
> > > > > >
> > > > > > I would like to recommend running on a non-gridworld environment since the major concern comes from the grid world. According to the submitted code, I think it is easy to incorporate Atari game environment (e.g., breakout, space invaders, SeaQuest, or other environments in Figure 2 in Burda et al 2019.)
> > > > > >
> > > > > > Burda, Yuri, et al. "Large-scale study of curiosity-driven learning." ICLR 2019.

---

> > > > > > > ### Author Response · Authors · 2022-11-19
> > > > > > > **Thanks for the suggestion**
> > > > > > >
> > > > > > > We thank the reviewer for the suggestion. We will try our best to run experiments on the suggested non-grid environments and update the results.

---

> > > > > > > ### Author Response · Authors · 2022-12-10
> > > > > > > **Updates on experiments**
> > > > > > >
> > > > > > > Dear Reviewer bLVL,
> > > > > > >
> > > > > > > Thank you very much for the suggestions and comments. In the last 2 weeks, we made our best efforts to run the experiments for non-gridworld environments. Here are the things we tried:
> > > > > > > * We notice that NovelD and RIDE also use episodic counts for complex environments such as Atari and Vizdoom. Therefore, we tried to run experiments on Atari and Vizdoom with their open-sourced code. However, we find that current NovelD's code only supports MiniGrid ([see here](https://github.com/tianjunz/NovelD/blob/ed9680e9ddb81e4b46a7020f03f6c483b06c1d8d/src/algos/bebold.py#L195)). For RIDE, we first encountered some environment wrapper bugs when running their code on Atari and Vizdoom. Even though we fixed the bugs, there were NaN issues and it was hard to pin down the cause.
> > > > > > > * We tried to implement the reachability network [2] in our codebase but it did not work even on MiniGrid. We tried a grid search for the hyper-parameters on MultiRoom-N7S8, e.g., learning rate, number of positive and negative pairs, training epochs, intrinsic reward multiplier. But we still have not obtained good results. The average return is almost always 0.
> > > > > > > * We notice that the ablation study in Figure 3 of the NGU paper [1] shows that episodic intrinsic rewards contribute more to the performance than lifelong intrinsic rewards in non-gridworld environments (Atari games). We tried to reproduce their results with [a third-party re-implementation](https://github.com/michaelnny/deep_rl_zoo). The experiments are still running and we will update the results in the final version. We believe the ablation experiments in NGU paper can corroborate our findings for non-gridworld environments.
> > > > > > >
> > > > > > > Since the discussion period is closing soon, we are wondering if your concerns have been addressed. If not, we would be happy to continue the discussion and/or revise the paper.
> > > > > > >
> > > > > > > [1] Badia, Adrià Puigdomènech, et al. "Never give up: Learning directed exploration strategies." arXiv preprint arXiv:2002.06038 (2020).
> > > > > > >
> > > > > > > [2] Savinov, Nikolay, et al. “Episodic curiosity through reachability.” ICLR. 2019.

---

> ### Author Response · Authors · 2022-11-18
> **Updated experiments results**
>
> As suggested by the reviewer, we have re-run the open-sourced code of NovelD on two tasks that are not reported in their paper (`MultiRoom-N10S10` and `MultiRoom-N12S10`), and included the results in Fig. 23 in the revised version. We managed to finish 3 runs before the deadline of discussion stage 1. We will do more runs and update the results before the end of discussion stage 2.
>
> Regarding comparison with the reachability network [1], we tried to implement their method for MiniGrid. We did a grid search for the hyper-parameters on `MultiRoom-N7S8`, e.g., learning rate, number of positive and negative pairs, training epochs, and intrinsic reward multiplier. But we have not obtained good results so far. The average return is 0. We will continue to try our best on this experiment.
>
> [1] Savinov, Nikolay, et al. "Episodic curiosity through reachability." ICLR. 2019.

---

### Official Review · Reviewer_e8Yz · 2022-10-24

**Confidence:** 5
**Correctness:** 4
**Technical Novelty And Significance:** 2
**Empirical Novelty And Significance:** 4
**Recommendation:** 8

**Clarity, Quality, Novelty And Reproducibility:**

Clarity:
- Medium-high. The paper is very clearly written and easy to understand. The main message is simple, but in my opinion important. I have some comments below that would improve the clarity.

Quality:
- Medium-high. The paper's experimental investigations are quite thorough, with many combinations of episodic and lifelong bonuses investigated. This shows that their conclusions are robust and not an artifact of just one setting.

Novelty:
- Medium-high. The methods are not very novel, but the insights are. I think this is important.

Reproducibility:
- High. The authors provide code to reproduce their experiments.


## Detailed comments:


- Section 2.2: When describing RIDE, it says: "Raileanu et al propose a novel lifelong curiosity". RIDE is actually not really a lifelong curiosity bonus - it measures the distance between consecutive embeddings, which will not disappear with sufficient exploration the same way RND, NovelD or ICM would (ICM will only vanish if the env is deterministic though). I would fix this in the text.
- Equation 7: NovelD/BeBold has a scaling constant in front of the second RND error, please add
- Section 3.1, in training details: it is mentioned that IMPALA is too slow and that PPO is faster, which is interesting since most works use IMPALA on MiniGrid. Please include a comparison of speed between the two, this could be useful to others.
- Throughout the paper, there are references to the Appendix but not specific parts. Please organize the appendix into sections and include references to the different parts, like (Appendix A.1, Appendix B.3, etc so people can easily find which what you're referring to in the main text).
- In Figures 2 and 3, which lifelong curiosity bonus is being used? Currently it only says "lifelong"
- In Section 3.3, it says the BeBold code is not open sourced, but it is now: https://github.com/tianjunz/NovelD. Although I'm guessing running the official code vs. the reimplemented code won't qualitatively change the results, please rerun using the official code for the camera ready.
- In Section 3.3, is the same underlying RL algorithm being used? I.e. RIDE, BeBold/NovelD use IMPALA but the count-based implementation uses PPO it seems. Please make sure the same underlying RL algorithm is the same, since this can make a big difference in sample complexity (IMPALA is in general less sample efficient than PPO since it only uses each sample once).
- In several places throughout the paper, there is both a \citet{} ref and a \citep{} ref - please only use one.

Minor grammar/language comments:
- Abstract: "only using lifelong curiosity can hardly make progress" -> "...hardly makes progress..."
- Intro: "recent works...pay increasing attention to" -> "recent works have paid increasing attention to"
- "does not cause significant performance drop" -> "does not cause a significant performance drop"
- "our work makes following contributions" -> "our work makes the following contributions"
- Section 2.1: ". Next state observation" -> ". The next state observation"
- Section 3.2: "agent chieves" -> "agent achieves"
- Section 3.2: "By contrast" -> "In contrast"
- Title 3.4: "Why lifelong curiosity does not help?" -> either "Why lifelong curiosity does not help" or "Why does lifelong curiosity not help?"
- Section 5: "skeptical to overfitting" -> "susceptible to overfitting"

**Strength And Weaknesses:**

Strengths:
- The paper's experimental study is very thorough. They examine a large number of combinations of episodic and lifelong novelty bonuses, and tune the intrinsic reward coefficient well, which can have a large impact on performance. They show that when this is well-tuned, the episodic bonus alone is sufficient to get state of the art performance.
- Although the paper does not propose a novel method, the main insight is novel and in my opinion high impact: it shows that to some extent, we have been thinking about exploration in procgen environments wrong, that lifelong novelty bonuses used for singleton RL environments do not automatically transfer over to procgen settings, and that episodic bonuses are more promising.

Weaknesses:
- While the paper shows that episodic bonuses are indeed helpful, the episodic bonuses it considers are all count-based which will only work in fairly simple settings like MiniGrid where there are not many distractors (which would break count-based bonuses). So although the paper does give compelling evidence that episodic bonuses deserve more investigation, it doesn't suggest a good or general solution based on episodic bonuses.


As a side note, this paper was posted after the ICLR submission deadline so does not affect the novelty of the present paper, but the authors might be interested in it since it is quite related: https://arxiv.org/abs/2210.05805. They also find that episodic bonuses are very important and propose a way generalize beyond count-based episodic bonuses.

**Summary Of The Paper:**

This paper conducts a thorough experimental study of episodic and lifelong curiosity bonuses for procedurally-generated (PG) environments. While several prior works have proposed different lifelong novelty bonuses with various motivations (RIDE, BeBold/NovelD, AGAC), and include episodic bonuses as a detail, this work instead shows that the episodic bonus is the main driver of performance, and that the lifelong novelty bonuses have little effect. This is an interesting insight which suggests rethinking exploration in procgen environments is needed.

**Summary Of The Review:**

Overall, I recommend this paper for acceptance because I believe the paper provides important insights for exploration in procgen environments. While there has been lots of work on exploration in singleton MDPs in the past, there has been increasing interest in procgen environments (and contextual MDPs more generally), and this paper shows that fundamentally different kinds of bonuses are needed for this setting (i.e. episodic rather than lifelong). For this reason, I think it's important to make the insights of this paper more widely known.

The paper does have some things that need fixing in the writing (as per my comments above), and a few changes experiment wise (i.e. rerunning BeBold experiments with offiical codebase which is now released), but I think these are relatively minor and can be done for the camera ready.

---

> ### Author Response · Authors · 2022-11-10
> **Authors' response (1/2)**
>
> Thank the reviewer for the constructive feedback. We are very glad to see the comments that experimental study is very thorough, and that the main insight is novel and has high impact. Below we respond to the comments. We also upload a revised version of the paper that incorporates the suggested changes.
>
> > `> Comment 1 (C1):` While the paper shows that episodic bonuses are indeed helpful, the episodic bonuses it considers are all count-based which will only work in fairly simple settings like MiniGrid where there are not many distractors (which would break count-based bonuses). So although the paper does give compelling evidence that episodic bonuses deserve more investigation, it doesn't suggest a good or general solution based on episodic bonuses.
>
> `> Response 1 (R1):` We agree that more experiments and analysis on more complex environments would further strengthen our paper. Our paper is more like a comprehensive empirical study on MiniGrid, which is extensively used in prior works (as the main benchmark) to study exploration in procedurally-generated environments. However, how episodic and lifelong curiosity contributes to the exploration performance on this important benchmark has not received much attention. Thus, we believe it is worthwhile sharing our observations with the community. In future works, we will explore how to design a better general episodic curiosity.
>
> > `> C2:` As a side note, this paper was posted after the ICLR submission deadline so does not affect the novelty of the present paper, but the authors might be interested in it since it is quite related: https://arxiv.org/abs/2210.05805. They also find that episodic bonuses are very important and propose a way generalize beyond count-based episodic bonuses.
>
> `> R2:` Yes, we also came across this paper after submission and it is indeed quite related. We have added it to the related works in the revised version.
>
> > `> C3:` Section 2.2: When describing RIDE, it says: "Raileanu et al propose a novel lifelong curiosity". RIDE is actually not really a lifelong curiosity bonus - it measures the distance between consecutive embeddings, which will not disappear with sufficient exploration the same way RND, NovelD or ICM would (ICM will only vanish if the env is deterministic though). I would fix this in the text.
>
> `> R3:` Thanks for the correction. We have revised it accordingly in the new version.
>
> > `> C4:` Equation 7: NovelD/BeBold has a scaling constant in front of the second RND error, please add
>
> `> R4:` Thank the reviewer for pointing it out. We notice that this scaling constant seems to be specific to NovelD. We will add it in the final version after we replace all occurrences of BeBold with NovelD.
>
> > `> C5:` Section 3.1, in training details: it is mentioned that IMPALA is too slow and that PPO is faster, which is interesting since most works use IMPALA on MiniGrid. Please include a comparison of speed between the two, this could be useful to others.
>
> `> R5:` We tried the open-sourced code of RIDE, which is based on an IMPALA implementation -- [TorchBeast](https://github.com/facebookresearch/torchbeast) (specifically the less scalable version, MonoBeast). The low efficiency is mainly because it uses CPU to do the forward pass of the policy during experience collection and it is non-trivial for us to migrate it to GPU. This also means that we need a large number of CPUs (typically 100+) to collect samples in parallel to reach the full speed. Such computation requirement (high CPU but low GPU demand) does not fit hardware resources in a university lab (where each workstation often has less than 32 CPUs). Based on our estimation, with the roughly same amount of resources, using PPO can be 10~20x faster. Note that we do not mean that one run of PPO is 10x faster than IMPALA. Rather, using PPO allows us to run multiple experiments in parallel under the same budget, and make full use of our hardware resources. We have included the above discussion in Appendix A.3.
>
> > `> C6:` Throughout the paper, there are references to the Appendix but not specific parts. Please organize the appendix into sections and include references to the different parts, like (Appendix A.1, Appendix B.3, etc so people can easily find which what you're referring to in the main text).
>
> `> R6:` Thank the reviewer for this good suggestion. We have fixed it in the revised version.

---

> > ### Author Response · Authors · 2022-11-10
> > **Authors' response (2/2)**
> >
> > > `> C7:` In Figures 2 and 3, which lifelong curiosity bonus is being used? Currently it only says "lifelong"
> >
> > `> R7:` For Figure 2, the results in each **row** use one kind of lifelong curiosity (as indicated by the label at the leftmost of each row). For Figure 3, the results in each **column** use one kind of lifelong curiosity. For example, the first row in Figure 2 compares the performance of $r^\textrm{ICM}$ only, $r^\textrm{ep\\_visit}$ only, $r^\textrm{ep\\_count}$ only, $r^\textrm{ICM}$ + $r^\textrm{ep\\_visit}$, $r^\textrm{ICM}$ + $r^\textrm{ep\\_count}$ and no intrinsic reward.
> >
> > > `> C8:` In Section 3.3, it says the BeBold code is not open sourced, but it is now: https://github.com/tianjunz/NovelD. Although I'm guessing running the official code vs. the reimplemented code won't qualitatively change the results, please rerun using the official code for the camera ready.
> >
> > `> R8:` Thank the review for the suggestion. We are currently rerunning it and will update the results soon.
> >
> > > `> C9:` In Section 3.3, is the same underlying RL algorithm being used? I.e. RIDE, BeBold/NovelD use IMPALA but the count-based implementation uses PPO it seems. Please make sure the same underlying RL algorithm is the same, since this can make a big difference in sample complexity (IMPALA is in general less sample efficient than PPO since it only uses each sample once).
> >
> > `> R9:` In Section 3.3, we aim to keep the results of other methods consistent with what is reported in their papers, by asking the authors for their training results or re-running their released code. Therefore, for BeBold and RIDE, IMPALA is the base RL algorithm. We already show in Section 3.2 that, with PPO as the same underlying RL algorithm, using episodic curiosity outperforms these two methods. Note that $r^\textrm{RIDE}$ + $r^\textrm{ep\\_count}$ and $r^\textrm{BeBold}$ + $r^\textrm{ep\\_visit}$ are respectively the full RIDE and BeBold methods. For RAPID and AGAC, they also use PPO as the base RL algorithm.
> >
> > > `> C10:` In several places throughout the paper, there is both a \citet{} ref and a \citep{} ref - please only use one.
> >
> > `> R10:` Thank the reviewer for pointing out this issue. We have fixed them in the revised version.
> >
> > > `> C11:` Minor grammar/language comments.
> >
> > `> R11:` Thank the reviewer for pointing them out. We have fixed them in the revised version.

---

> > > ### Comment · Reviewer_e8Yz · 2022-11-18
> > > **Thank you for the updates**
> > >
> > > Thank you for the updates, I've kept my recommendation of accept. This is a nice paper.

---

> > > > ### Author Response · Authors · 2022-11-18
> > > > **Thank you!**
> > > >
> > > > Thank you very much for the comments and feedback! We are very glad that you like our work.

---

> ### Author Response · Authors · 2022-11-18
> **Updated experiments results of re-running NovelD's code**
>
> As suggested by the reviewer, we have re-run the open-sourced code of NovelD on two tasks that are not reported in their paper (`MultiRoom-N10S10` and `MultiRoom-N12S10`), and included the results in Fig. 23 in the revised version. We managed to finish 3 runs before the deadline of discussion stage 1. We will do more runs and update the results before the end of discussion stage 2.

---

### Official Review · Reviewer_GnK2 · 2022-10-26

**Confidence:** 3
**Correctness:** 2
**Technical Novelty And Significance:** 2
**Empirical Novelty And Significance:** 2
**Recommendation:** 3

**Clarity, Quality, Novelty And Reproducibility:**

Novelty: My understanding is that each episode is a new layout of the task. In this way, each episode presents a new exploration problem to tackle: the keys, doors, obstructions, and goals have all moved around. To be successful, the agent needs to explore thoroughly once again in every episode and cannot rely much on what it has previously learned about the value of each state. For this reason, it does not seem especially novel to note that an intrinsic reward that encourages complete re-exploration of the environment in each episode would perform better than an intrinsic reward that assumes that parts of the world that the agent visited in previous episodes are no longer novel enough to be worth exploring—the agent needs to re-explore in each episode!

That said, as the authors of this paper have noted, prior work, even those working with MiniGrid, has sometimes focused on these lifelong intrinsic reward designs. This suggests that the core idea of this work is not completely obvious, so seems worth sharing.

Clarity: Much of the paper is quite clear, but there are some areas where the writing is vague or nonspecific on details to the point that the paper becomes difficult to follow.
- As you've written the section on ICM (p. 3), it sounds like the inverse model has no effect on the ICM intrinsic reward, so it sounds like extraneous information. It should be made clear how the inverse model plays into the ICM intrinsic reward.
- "is too time-consuming" (p. 4) ← Do you mean in terms of ease of implementation, training time, or something else?
- "curiosity about the states" (p. 2) ← given that the intrinsic reward is computed from a transition tuple, does the intrinsic reward sometimes capture something about actions too (not just states)?

Reproducibility: Section 3.4 is especially problematic for reproducibility because it lacks many details that are important for understanding, nevermind attempts to replicate the experiments.
1. "after training it" (p. 7) ← How much training has the agent received? Have you frozen the intrinsic reward or the value function, or is the agent still learning?
2. "randomly permuting the lifelong curiosity within a batch" (p. 2), "we randomly permute the lifelong curiosity within a batch" (p. 7) ← It is unclear how this permutation is performed.
3. "We first generate trajectories that accomplish the tasks and cover all rooms." (p. 7) ← How are these trajectories generated?

There are also a number of other details missing:
1. "Surprisingly, we find that only using episodic curiosity, the trained agent can match or surpass the performance of the one trained by using lifelong-episodic combination; in contrast, only using lifelong curiosity makes little progress in exploration." (p. 1) ← You refer to "performance" generally here, but at this point you have also implied two different performance metrics: one being extrinsic return and the other being some kind of exploration metric when no extrinsic reward is provided to the agent. Can you be a little more explicit? (The same general use of the word "performance" also continues onto the next page.)
2. "the learned lifelong curiosity assigns similar values to both novel and familiar states." (p. 2) You say "learned" here; is this based on a measurement after some number of episodes? How are novel and familiar defined here?
3. "When applicable" (p. 5) → Can you be more specific about when this statement is and is not applicable? Noting that some of the results are taken from other papers, not being explicit calls into question whether or not the comparisons in the figure are reasonable.
4. Please define what a run is. ("When applicable, the training curves are averaged over 5 runs and the standard deviations are plotted as shaded areas." p. 5)
5. "the average number of explored rooms within an episode" (p. 5) Please define what an "explored room" is (does the agent have to visit every grid square in the room and toggle all the objects? Visit the doorway? Visit any grid square of the room? Can a room be partially explored or is this a discrete metric?

**Strength And Weaknesses:**

This paper is well-organized. The experiments seem to provide a reasonable demonstration of the key point: in MiniGrid, episodic intrinsic reward bonuses appear provide more benefit in terms of extrinsic performance and exploration coverage than lifelong intrinsic reward bonuses.

However, the descriptions of the experiments and results are not completely clear and seem to be missing some key details (as detailed in my answer on Clarity and Reproducibility).

There are a few errors/inaccuracies that don't seem to form a larger pattern, but should be addressed in revision of this paper:
1. "This demonstrates that the learned lifelong curiosity does not accurately assign high values to novel states and low values to states the agent is familiar with." (p. 7) ← I don't think this conclusion logically follows from the evidence provided.
2. "A possible reason could be that the learned lifelong curiosity is not able to accurately reflect the novelty of states, making the agent receive higher cumulative (intrinsic) rewards from the opening-closing behavior than from exploring new rooms." (p. 7) ← I found this statement concerning, because it appears to reflect either a misunderstanding about intrinsic reward methods or a miscommunication where I am misunderstanding what the authors are trying to say. Whether an agent would receive higher rewards by exploring new rooms than opening and closing the door is sort of beside the point: the agent has never experienced the rooms it has never visited before, so it doesn't have any estimate for the value of new rooms, so there is nothing to direct its behaviour towards those rooms. The only thing that matters is whether the opening-closing states have higher value than the states the agent nearby that the agent has visited before. This is a known problem with exploration reward bonuses. This is related to what Shyam et al. (2019) refer to as reactivity or what Ecoffet et al. (2021) refer to as detachment.  Intrinsic rewards only influence behaviour with respect to previously visited states, so methods that rely on them also have to rely on stochastic behaviour or careful search control on exploration frontiers.

Some typos, grammar, and concision suggestions (just to help with editing, no need to respond to these):
- "environments draws increasing" (p. 1) → "environments has drawn increasing"
- "This finding would be inspiring for" (p. 2) → consider something like "This finding should inspire"
- "to refer to the corresponding reward as a whole rather than the reward at timestep t." (p. 2) ← This choice may leave open some confusion, given that r is also defined as the reward function itself in Section 2.1.
- "Raileanu et al. (Raileanu & Rocktäschel, 2020)" (p. 3, appears in both RIDE and Episodic visitation count sections) → "Raileanu & Rocktäschel (2020)"
- "Zhang et al. (Zhang et al., 2020)" (p. 3, appears in both BeBold and Episodic first visits sections) → "Zhang et al. (2020)"
- "encourages the agent to explore beyond the boundary of explored regions." (p. 3) → without all of the intuition provided by Zhang et al., it's not very clear what is means to explore beyond the boundary of explored regions, so you might do better to provide intuition about the structure of the reward.
- Once the lifelong component of RIDE is removed, the episodic visitation count bonus reduces to an MBIE-EB-style bonus, which is quite well-studied, so it would be good to include a reference to Strehl & Littman (2008) to help the reader find more information.
- "sparse extrinsic reward on accomplishing the task and pure exploration without extrinsic reward (Section 3.2)." (p. 4) ← This phrase is quite difficult to parse, so I would recommend splitting it up into multiple parts, something like "(a) a sparse intrinsic reward setting where reward is only obtained upon accomplishing the task, and (b) a pure exploration setting without extrinsic rewards."
- "door in order to enter next room." (p. 4) --> "door to enter the next room." While "in order" is not incorrect, it lengthens the sentence unnecessarily.
- "grid as observation and choose from 7 actions" --> "grid as the observation and chooses from 7 actions" (p. 4)
- It would be helpful to have some explanation of what the non-obvious 'done' action does. (p. 4)
- "The agent obtains a non-zero reward" (p. 4) ← "In our sparse reward setting, the agent obtains a non-zero reward" because I assume that's not true in the pure exploration setting of your experiments.
- "deferred to Appendix." --> "deferred to Appendix A.1." (p. 4)
- "are included in Appendix." → "are included in Appendix A.4." (p. 4)
- "(see Eqn. equation 2)." → "(see Equation 2)." (p. 5)
- "have large impact" → "have a large impact" (p. 3, p. 5)
- "Detailed configurations about hyperparameters are included in Appendix." (p. 5) ← I'm not sure what configurations about hyperparameters are; it might be more useful to say that a detailed explanation of how the hyperparameter search was performed and demonstrations of sensitivity to different values of beta can be found in Appendix A.5.
- "Here “None” refers to not using lifelong or episodic curiosity. If both are not used, then it reduces to vanilla PPO without intrinsic rewards." (p. 5) ← And if only one is used, I assume the other reduces to 1, because they are combined multiplicatively?
- "chieves" (p. 5) → "achieves"
- "when agents reaching the goal." → "when agents reach the goal" or "when an agent reaches the goal" (p. 5)
- "the agent starts from the first room" (p. 5) ← Is there some way of picking out the first room besides it being where the agent starts? This statement seems tautological.
- In the caption to Figure 4, it would be great to reiterate "the performance of BeBold in ObstructedMaze-2Q is directly taken from their paper but we cannot reproduce it," because it sticks out as an outlier amongst your results.
- "recent proposed ones" (p. 6) → "recently proposed ones"
- "Such gap" (p. 6) → "This difference in exploration efficiency"
- "included in Appendix" (p. 7) → "included in Appendix A.4"
- "In Appendix, we also include the comparison results on other environments in MiniGrid benchmark." (p. 7) → "In Appendix A.6, we also include the comparison results on other environments in MiniGrid."
- "3.4 Why lifelong curiosity does not help?" (p. 7) → "3.4 Why doesn't lifelong curiosity help?"
- "as shown in Figure 6 is" (p. 7) → "as shown in Figure 6, is"
- "Bebold" (p. 12) → "BeBold"
- "comarison" (p. 19) → "comparison"

Ecoffet, A., Huizinga, J., Lehman, J., Stanley, K. O., & Clune, J. (2021). First return, then explore. Nature, 590(7847), 580-586.

Shyam, P., Jaśkowski, W., & Gomez, F. (2019, May). Model-based active exploration. In International Conference on Machine Learning (pp. 5779-5788). PMLR.

Strehl, A. L. and Littman, M. L. (2008). An analysis of model-based interval estimation for Markov decision processes. Journal of Computer and System Sciences, 74(8):1309 – 1331.

**Summary Of The Paper:**

In some papers using intrinsic reward methods, the intrinsic reward is composed of two components: lifelong intrinsic reward (where the "novelty" of each state is tracked across multiple episodes) and episodic intrinsic reward (where the "novelty" of each state is tracked only across the current episode). The core idea of this paper is that, within the MiniGrid benchmark task, lifelong intrinsic rewards contribute little to agent performance, while episodic intrinsic rewards provide the bulk of performance improvements. The authors provide evidence for these differences in contributions via a battery of experiments on MiniGrid, replicating and ablating those components of several recent intrinsic reward proposals.

**Summary Of The Review:**

I am leaning to reject this paper because it is missing too many details, to the point that it is somewhat difficult to follow and assess. I think that the primary message of the paper is sufficiently well-supported for acceptance, but the paper as a whole is not in acceptable form. I recommend the authors clear up the errors and missing details in the paper to bring it to the acceptance threshold.

---

> ### Author Response · Authors · 2022-11-10
> **Authors' response (1/2)**
>
> Thank the reviewer for the constructive feedback and the very detailed comments. We are glad to see the comments that our paper is well-organized, and that the primary message of our paper is sufficiently well-supported for acceptance. Below we respond to the comments and provide the details missing in the first version. We also upload a revised version of the paper that incorporates the suggested changes.
>
> * **Weaknesses**
>     > `> Comment 1 (C1):` There are a few errors/inaccuracies that don't seem to form a larger pattern, but should be addressed in revision of this paper.
>
>    `> Response 1 (R1):` Thank the reviewer for pointing them out. We have revised the corresponding parts and would like to hear further feedback.
>     > `> C2:` Some typos, grammar, and concision suggestions (just to help with editing, no need to respond to these).
>
>     `> R2:` We really appreciate the reviewer's time in going through our paper so carefully. We have revised our paper as suggested.
>
> * **Novelty**
>
>   We agree that it is not surprising that in procedurally-generated environment the agent needs to explore thoroughly in each novel episodes. As the reviewer puts it, our intention is to share with the community that the contribution of episodic curiosity is largely overlooked in previous works.
>
> * **Clarity**
>     > `> C1:` As you've written the section on ICM (p. 3), it sounds like the inverse model has no effect on the ICM intrinsic reward, so it sounds like extraneous information. It should be made clear how the inverse model plays into the ICM intrinsic reward.
>
>     `> R1:` In the revised version, we remove extraneous descriptions of the inverse model to keep it concise.
>     > `> C2:` "is too time-consuming" (p. 4) ← Do you mean in terms of ease of implementation, training time, or something else?
>
>     `> R2:` We mean the training time is too long. We have clarified it in the new version.
>     > `> C3:` "curiosity about the states" (p. 2) ← given that the intrinsic reward is computed from a transition tuple, does the intrinsic reward sometimes capture something about actions too (not just states)?
>
>     `> R3:` Yes, it depends on how the intrinsic reward is calculated. For the 4 previous methods introduced in Sec. 2.2, they mainly use $s_t$ and $s_{t+1}$ to define the intrinsic reward while ignoring $a_t$. But in principle, we can make it depend on $a_t$. We only mention "curiosity about the states" to keep consistency with the context "Following previous curiosity-driven approaches".
>
> * **Reproducibility**
>     > `> C1:` "after training it" (p. 7) ← How much training has the agent received? Have you frozen the intrinsic reward or the value function, or is the agent still learning?
>
>     `> R1:` The agent is trained for 2e7 frames under the "lifelong only" setting, i.e., we take the trained agent in the experiments in Fig. 2 (dark green lines). The whole policy is frozen and we simply roll out the learned policy in the environment to examine the agent's behavior. We make clarifications in the revised version.
>     > `> C2:` "randomly permuting the lifelong curiosity within a batch" (p. 2), "we randomly permute the lifelong curiosity within a batch" (p. 7) ← It is unclear how this permutation is performed.
>
>     `> R2:` Let $\\{(s_j, a_j, s_j')\\}^B_{j=1}$ denote a batch of transitions and $\\{r_j\\}^B_{j=1}$ denote the corresponding batch of computed lifelong curiosities. We randomly permute the curiosities in the batch $\\{r_j\\}^B_{j=1}$, such that each transition $(s_j, a_j, s_j')$ might be associated with the wrong curiosity $r_k$ (where $k\neq j$) rather than the correct one. We have added clarifications in the revised version.
>     > `> C3:` "We first generate trajectories that accomplish the tasks and cover all rooms." (p. 7) ← How are these trajectories generated?
>
>     `> R3:` These trajectories are collected by a trained agent that achieves high reward in the task (e.g., one trained with "ep_visit only"). We have updated the paper to clarify it.

---

> > ### Author Response · Authors · 2022-11-10
> > **Authors' response (2/2)**
> >
> > * **Missing details**
> >     > `> C1:` "Surprisingly, we find that only using episodic curiosity, the trained agent can match or surpass the performance of the one trained by using lifelong-episodic combination; in contrast, only using lifelong curiosity makes little progress in exploration." (p. 1) ← You refer to "performance" generally here, but at this point you have also implied two different performance metrics: one being extrinsic return and the other being some kind of exploration metric when no extrinsic reward is provided to the agent. Can you be a little more explicit? (The same general use of the word "performance" also continues onto the next page.)
> >
> >     `> R1:` Thanks very much for pointing out this unclear description. Your understanding is right. We have revised it and made the performance metrics explicit.
> >     > `> C2:` "the learned lifelong curiosity assigns similar values to both novel and familiar states." (p. 2) You say "learned" here; is this based on a measurement after some number of episodes? How are novel and familiar defined here?
> >
> >     `> R2:` "Learned" refers to the lifelong curiosity module after training (e.g., 4e7 steps for `KeyCorridor-S4R3` environment). Here "novel" and "familiar" refer to the states where the agent has few or many visitations. Since it is in the introduction section, we mainly focus on high-level intuitions while deferring the details to the experiment section.
> >     > `> C3:` "When applicable" (p. 5) → Can you be more specific about when this statement is and is not applicable? Noting that some of the results are taken from other papers, not being explicit calls into question whether or not the comparisons in the figure are reasonable.
> >
> >     `> R3:` For the experiments conducted by us, the results are averaged from 5 runs. For the results directly taken from previous papers, they are not necessarily averaged from 5 runs. Appx. A.4 details which results are directly taken from their paper or not. Considering it causes more confusion, we have removed "When applicable" in the revised version.
> >     > `> C4:` Please define what a run is. ("When applicable, the training curves are averaged over 5 runs and the standard deviations are plotted as shaded areas." p. 5)
> >
> >     `> R4:` A run is a full training process with a random seed for a specific experiment configuration, e.g., training 4e7 steps in the `KeyCorridor-S4R3` environment under "ep_visit only" setup.
> >     > `> C5:` "the average number of explored rooms within an episode" (p. 5) Please define what an "explored room" is (does the agent have to visit every grid square in the room and toggle all the objects? Visit the doorway? Visit any grid square of the room? Can a room be partially explored or is this a discrete metric?
> >
> >     `> R5:` Thank the reviewer for spotting such unclarity. In our experiments, we consider a room is explored if the agent visit any square in this room, thus it is a discrete metric. We have added clarifications in the new version.

---

> ### Comment · Reviewer_GnK2 · 2022-11-18
> **More editing**
>
> You've made some helpful changes; thanks! Here are some comments and suggestions to sink your teeth into while I continue to re-review.
>
> I strongly recommend using the term "intrinsic reward" over "curiosity" in most parts of the paper. "Intrinsic reward" aligns more closely with other literature studying intrinsic rewards generally.
>
> Minor issues:
> - "In MiniGrid, the agent receives a partial view of the grid" (p. 4) ← I believe MiniGrid offers a "fully observable" option; for this reason, I think you should write this phrase to make it clear that you are choosing partial view as an option, not that it is what always occurs in MiniGrid.
> - I'm still unclear about which moment a room counts as "visited;" does visiting the tile with the door count as visiting a tile in the room?
> - "or the number of training frames, we follow the same configuration in (Zhang et al., 2020)." (p. 4) → This feels like a literature goose chase since Zhang et al. then send the reader to Raileanu and Rocktäschel ("we follow the exact training paradigm from (Raileanu and Rocktäschel, 2020)." p. 5)
>
> Typos/Sentence-level suggestions:
> - "2.1 Notations" (p. 2) → "2.1 Notation"
> - "which encourage the agent" (p. 3) → "which encourages the agent"
> - "door to enter next room." (p. 4) → "door to enter the next room."
> - "follow the same configuration in (Zhang et al., 2020)." (p. 4) → "use the same configuration as Zhang et al. (2020)." or "follow the configuration used by Zhang et al. (2020)."
> - "following the same procedure in (Raileanu & Rocktäschel, 2020)." (p. 5) → "following the same procedure as that used by Raileanu & Rocktäschel (2020)." or "following the procedure used by Raileanu & Rocktäschel (2020)."
> - "the it reduces to 1" (p. 5) → "then the reward reduces to 1"
> - "without providing the extrinsic reward when agents reach the goal." → "without providing any extrinsic reward." (Otherwise, it sounds like you might provide other extrinsic rewards, just not the one for reaching the goal.)
> - "directly taken from their paper." (p. 6) → I recommend something like "directly taken from the associated paper (Zhang et al., 2020)."
> - "we contact the authors and obtain the original" (p. 7) → "we contacted the authors and obtained the original"
> - "we obtain results" (p. 15) → "we obtained results"

---

> > ### Author Response · Authors · 2022-11-19
> > **Thank you for the feedback**
> >
> > Thank you very much for providing detailed comments. We believe our paper has been greatly improved by your suggestions.
> >
> > > I strongly recommend using the term "intrinsic reward" over "curiosity" in most parts of the paper. "Intrinsic reward" aligns more closely with other literature studying intrinsic rewards generally.
> >
> > We have revised our paper to replace "curiosity" with "intrinsic reward", including the title. Thank you for the suggestion.
> >
> > > Does visiting the tile with the door count as visiting a tile in the room?
> >
> > In our implementation, we do not count this case as visiting a tile in the room. We make it clear in the revised version.
> >
> > > This feels like a literature goose chase
> >
> > Fig. 12 in Appendix A.1 summarizes the number of training frames for each task, and is meant to stop the goose chase. We have revised the main text to explicitly mention Fig. 12.
> >
> > > Other minor issues, typos and sentence-level suggestions
> >
> > We would like to express our deep thanks to you for carefully going through our paper. We have fixed them in the revised version.

---

> > > ### Comment · Reviewer_GnK2 · 2022-11-20
> > > **More editing plus some more substantive concerns**
> > >
> > > This paper is looking better and better with each revision, which is fantastic. I think the main remaining barrier to feeling that this paper is ready for publication is still that some claims seem under-supported. However, I do suspect that with a little more back-and-forth, we can resolve those so the paper can be accepted. Again, here are some comments so you don't have to wait for me to completely finish re-reviewing before getting started on them.
> > >
> > > - I agree with the other reviewers that the title should probably reflect that this work focuses on procedurally-generated sparse-reward grid worlds, rather than procedurally-generated environments in general. I may have missed another comment explaining why you don't feel such a change is inappropriate, sorry for asking you to reiterate, if so.
> > > - I don't think this sentence is logical: "This shows that the learned lifelong curiosity does not accurately reflect the novelty of states, otherwise such permutation would lead to obvious performance drop." (p. 7) A measure could perfectly accurately reflect the novelty of a state, yet using it as a reward signal in the particular algorithm of choice could still result in poor performance. Novelty accuracy does not imply better performance.
> > > - I'm uneasy about the statements that seem to directly credit (or blame) intrinsic rewards for behavior (mostly in Section 3.4). It seems like more attention should be paid to the learned value function. In most RL algorithms, the intrinsic reward associated with a single transition doesn't have much effect. I expect the value function to be what matters in these examples like the agent stuck with the door (p. 7) or the agent lingering in room 1 (p. 8). If Fig. 5 really does show the mean and max intrinsic reward associated with transitions within each room, rather than the mean and max of the value function (which I would expect to more directly influence agent behavior), it's hard for me to understand what this demonstrates. Please correct me if this is due to an intricacy of IMPALA—though including some explanation to help readers without detailed knowledge of IMPALA would probably be welcome, to speak to a larger subset of our community.
> > >
> > > Minor:
> > > - Why is Fig. 7 after Fig. 5, even though you discuss Fig. 7 before Fig. 5 in the text?
> > > - "An alternative solution" (p. 8) ← This solution is an "alternative" to what? But the sentence might be okay starting with something like "One possible solution," assuming I'm understanding this part correctly.
> > > - Are you using the phrase "cover all rooms" (p. 8) the same way you referred to "explored rooms," (p. 6) where "cover" means visit at least one tile of the room? I'd recommend using the "explored" term as you defined it earlier to reduce ambiguity since _coverage_ has connotations of the possibility that it could mean visiting every tile.
> > > - "quality of the learned latent embedding model" (pp. 8-9) ← Do you believe that this has to be a "learned latent embedding model" for these methods to work? I'm wondering if "quality of the representation" might more effectively make your point.
> > > - "the pseudo-counts of states derived from a density model (Bellemare et al., 2016; Ostrovski et al., 2017)" The whole point of pseudo-counts is to mimic anything you could do with counts in tabular state spaces. So the pseudo-counts themselves don't measure the difference between states: functions of counts/pseudo-counts (like the ones you explore in this paper) can be used for that. Is there any reason this sentence doesn't include "counts" as well?
> > >
> > > Typos/Sentence-level suggestions:
> > > - You use both "grid world" and "gridworld" and may want to pick one to use consistently
> > > - ”moving forward” and ”opening-door” (p. 7) have backward quotations on the front
> > > - "In our experiments, gridworld is used as the testbed, in which the visitation count of a state can be easily obtained as it has discrete state spaces." (p. 8) ← A bit convoluted. Maybe something like, "Our experiments use gridworld testbeds, which have discrete state spaces, so it is easy to obtain visitation counts for each state."
> > > - "but become incapable" (p. 8) ← mostly grammatical, but my recommendation would be "but performs poorly" instead.
> > > - "absence of the extrinsic" (p. 9) → "absence of extrinsic"
> > > - "next state Pathak et al. (2017) or a random target Burda et al. (2019b)." (p. 9) → "next state (Pathak et al., 2017) or a random target (Burda et al., 2019b)." This sentence could be further improved by reminding the reader that these two suggestions are ICM and RND as are explored throughout the paper, to help the reader make that connection.
> > > - "works under procedurally-generated" (p. 9) ← consider "using," "focused on," or "exploring" instead of "under"

---

> > > > ### Author Response · Authors · 2022-11-20
> > > > **Thank you for the editing! (1/2)**
> > > >
> > > > We greatly appreciate the reviewer's time and efforts in improving our paper to make it ready for publication. We have incorporated the suggested changes into our paper (but currently the submission cannot be updated during discussion stage 2).
> > > >
> > > > > I agree with the other reviewers that the title should probably reflect that this work focuses on procedurally-generated sparse-reward grid worlds, rather than procedurally-generated environments in general. I may have missed another comment explaining why you don't feel such a change is inappropriate, sorry for asking you to reiterate, if so.
> > > >
> > > >   As suggested by Reviewer pHUd, we have revised the main text to explicitly mention "gridworld". We can also modify the title to "procedurally-generated gridworlds" if the reviewer finds it necessary.
> > > >
> > > > > I don't think this sentence is logical: "This shows that the learned lifelong curiosity does not accurately reflect the novelty of states, otherwise such permutation would lead to obvious performance drop." (p. 7) A measure could perfectly accurately reflect the novelty of a state, yet using it as a reward signal in the particular algorithm of choice could still result in poor performance. Novelty accuracy does not imply better performance.
> > > >
> > > >   We agree with the reviewer's thinking that novelty accuracy does not imply better performance. We believe the experiments here at least show that the learned lifelong intrinsic reward is not critical to the exploration performance. The little change in performance after permutation indicates that the learned lifelong intrinsic reward functions as nothing more than a global scaling factor. We will update the sentence accordingly and would like to know if the reviewer think the new explanation is logical.
> > > >
> > > > > I'm uneasy about the statements that seem to directly credit (or blame) intrinsic rewards for behavior (mostly in Section 3.4). It seems like more attention should be paid to the learned value function. In most RL algorithms, the intrinsic reward associated with a single transition doesn't have much effect. I expect the value function to be what matters in these examples like the agent stuck with the door (p. 7) or the agent lingering in room 1 (p. 8). If Fig. 5 really does show the mean and max intrinsic reward associated with transitions within each room, rather than the mean and max of the value function (which I would expect to more directly influence agent behavior), it's hard for me to understand what this demonstrates. Please correct me if this is due to an intricacy of IMPALA—though including some explanation to help readers without detailed knowledge of IMPALA would probably be welcome, to speak to a larger subset of our community.
> > > >
> > > >   For actor-critic type algorithms (e.g., PPO and IMPALA), we think the behavior is more directly related to the parametric policy rather than the learned value function. Both the value and the reward do not immediately affect the behavior. They gradually influence the behavior during the training process through the advantage term in the policy gradient objective. Although the intrinsic reward associated with a single transition may not have much effect, the reward in different state-action will collectively affect the agent's behavior, since the objective is to maximize the expected cumulative rewards. If the learned lifelong intrinsic rewards are similar in room 1 and room 2, trajectories where the agent travels from room 1 to room 2 will not have higher return than trajectories where the agent stays in room 1. Fig. 5 is meant to provide a possible explanation on why the agent trained with "lifelong only" does not explore well. We will make it more clear and would also like to hear the reviewer's suggestion.

---

> > > > > ### Author Response · Authors · 2022-11-20
> > > > > **Thank you for the editing! (2/2)**
> > > > >
> > > > > **Minor:**
> > > > > * > Why is Fig. 7 after Fig. 5, even though you discuss Fig. 7 before Fig. 5 in the text?
> > > > >
> > > > >   Thank the reviewer for pointing out this layout issue. We have swapped the places of these two figures and it is consistent now.
> > > > >
> > > > > * > "An alternative solution" (p. 8) ← This solution is an "alternative" to what? But the sentence might be okay starting with something like "One possible solution," assuming I'm understanding this part correctly.
> > > > >
> > > > >   You understand it correctly. We have fixed it as suggested.
> > > > >
> > > > > * > Are you using the phrase "cover all rooms" (p. 8) the same way you referred to "explored rooms," (p. 6) where "cover" means visit at least one tile of the room? I'd recommend using the "explored" term as you defined it earlier to reduce ambiguity since coverage has connotations of the possibility that it could mean visiting every tile.
> > > > >
> > > > >   Thank you for the suggestion. We have changed it to "explore all rooms".
> > > > >
> > > > > * > "quality of the learned latent embedding model" (pp. 8-9) ← Do you believe that this has to be a "learned latent embedding model" for these methods to work? I'm wondering if "quality of the representation" might more effectively make your point.
> > > > >
> > > > >   We agree that "quality of the representation" better fits the context and have revised that sentence.
> > > > >
> > > > > * > "the pseudo-counts of states derived from a density model (Bellemare et al., 2016; Ostrovski et al., 2017)" The whole point of pseudo-counts is to mimic anything you could do with counts in tabular state spaces. So the pseudo-counts themselves don't measure the difference between states: functions of counts/pseudo-counts (like the ones you explore in this paper) can be used for that. Is there any reason this sentence doesn't include "counts" as well?
> > > > >
> > > > >   Thank the reviewer for pointing it out. We have changed it to "Such difference can be measured by the inverse square root of state counts or pseudo-counts (Bellemare et al., 2016; Ostrovski et al., 2017)".
> > > > >
> > > > > **Typos/Sentence-level suggestions:**
> > > > >
> > > > > Thank the reviewer very much for pointing them out. We have fixed them as suggested.

---

> > > > > > ### Comment · Reviewer_GnK2 · 2022-11-24
> > > > > > **Substantial concerns about Section 3.4**
> > > > > >
> > > > > > I still have fairly substantial concerns about the experimental philosophy and interpretation of the experiments in Section 3.4. The conclusions of this entire section should at least be reframed as highly speculative, since the experimental choices do not provide definitive evidence for the claims made.
> > > > > >
> > > > > > ### Static policy roll-out experiment
> > > > > > Experiment of interest: "we examine the learned behavior of an agent trained with the lifelong intrinsic rewards only (see Fig. 2), by rolling out the learned policy in the environment."
> > > > > >
> > > > > > The premise of this experiment is questionable. *Intrinsic rewards are not meant to determine a static policy; generally, they are better thought of as shaping temporary policies that ensure the agent visits each state enough times.*
> > > > > >
> > > > > > The behaviour described (e.g. "oscillating between two states") is exactly what I would expect from running one of these policies, so this section is not a helpful argument against lifelong intrinsic reward. Not something you'll be able to address on this timeline, but I would be curious about comparisons with the episodic intrinsic reward agents. I would be a bit surprised if ep_visit doesn't sometimes result in through looping or oscillatory policies.
> > > > > >
> > > > > > I want to return to my concern about the observation "but lifelong intrinsic reward it receives from this step is lower than previous ”opening-door” step. Thus, … the agent gets stuck." This phrasing makes it sound like a particular reward in a particular transition is the cause of the oscillating behaviour. This doesn't account for the way the policy is shaped iteratively over time, where one reward has relatively little effect, so the oscillating behaviour is probably not caused by just one random sample.
> > > > > >
> > > > > > ### Permutation experiment
> > > > > > Permuting over the transitions of a batch has the effect of averaging the intrinsic reward over “nearby” or “recently visited” states (those included in the batch). These averaged rewards still give the agent valuable information about the average novelty of the states, which depends on its behaviour.
> > > > > >
> > > > > > "We believe the experiments here at least show that the learned lifelong intrinsic reward is not critical to the exploration performance." (a) If we saw a big negative effect from permuting the rewards, then we could conclude that intrinsic rewards were useful, but not seeing such a big change doesn’t let us conclude that the intrinsic rewards are not useful. It could simply mean that permuting the rewards doesn’t destroy their usefulness. (b) I think you should be careful about calling this "exploration performance" because the y-axis of Figure 4 is return, and these two quantities shouldn’t be conflated. It might be more accurate to say that the lifelong intrinsic reward does not consistently contribute positively to performance.
> > > > > >
> > > > > > "The little change in performance after permutation indicates that the learned lifelong intrinsic reward functions as nothing more than a global scaling factor." Looking at Figure 4, in some combinations, the difference in the learning curves is actually pretty substantial (e.g. BeBold + ep_count) suggesting that the intrinsic reward results in very different behaviour than its permutations.
> > > > > >
> > > > > > ### Fig. 5 concerns
> > > > > > The time-varying nature of positive intrinsic rewards is key to how they work; so removing the temporal component via your choice of summary statistics (max, mean) removes any useful information from Figure 5. The mean and max statistics of intrinsic rewards for different states do not reflect the shape of the intrinsic reward distribution at any particular point in time, which is what matters in shaping policies. The important thing is that the intrinsic reward of room 1 will decay as long as the agent stays in it, while at the same time the intrinsic reward of room 2 remains high, eventually encouraging the agent to move forward. This is not reflected in the max and mean rewards while in each room, which could well be the same for the two rooms. Figure 5 can't support any of the claims made in the paper.
> > > > > >
> > > > > > ### Other
> > > > > > - Calling the lifelong intrinsic rewards "learned" is misleading; they are computed from other learned quantities, unless I'm mistaken, so the word "computed" might be a good substitute.
> > > > > > - I don’t understand what you mean by “Then for transitions in each trajectory, we calculate the learned lifelong intrinsic reward using models trained with r^{lifelong} only”. The only objects in your paper that could be called models and that take reward as input are RL agents.

---

> > > > > > > ### Author Response · Authors · 2022-11-26
> > > > > > > **Thank you for your helpful comments!**
> > > > > > >
> > > > > > > Thank you for sharing your concerns about Sec. 3.4. We will take the reviewer's suggestions and reframe Sec. 3.4 accordingly. We will make it clear that Sec. 3.4 contains just our speculations on why lifelong curiosity does not work, far from a solid investigation.
> > > > > > >
> > > > > > > > I would be curious about comparisons with the episodic intrinsic reward agents.
> > > > > > >
> > > > > > > As requested, we visualize the behaviour of an agent trained with ep_visit. Please see the following anonymized GIFs (the levels are randomly chosen):
> > > > > > > * [anonymized GIF 1](https://i.imgur.com/Tsq9l6D.gif)
> > > > > > > * [anonymized GIF 2](https://i.imgur.com/eAFqD3F.gif)
> > > > > > > * [anonymized GIF 3](https://i.imgur.com/phx11Ya.gif)
> > > > > > >
> > > > > > >
> > > > > > > We can see that the agent can explore all rooms without oscillating.
> > > > > > >
> > > > > > > > These averaged rewards still give the agent valuable information about the average novelty of the states, which depends on its behaviour.
> > > > > > > > (a) If we saw a big negative effect from permuting the rewards, then we could conclude that intrinsic rewards were useful, but not seeing such a big change doesn’t let us conclude that the intrinsic rewards are not useful. It could simply mean that permuting the rewards doesn’t destroy their usefulness.
> > > > > > >
> > > > > > > We agree that permutation has the effect of averaging the novelties of recent states (in a batch), and that not seeing such a big change after permutation could simply mean that permuting the rewards doesn’t destroy their usefulness. However, comparing the results in Figure 3 and Figure 7, we can see that permuting lifelong curiosity performs no better than simply using a constant (i.e., episode curiosity only). If the permuted lifelong curiosities are still useful in the averaging sense, then it should perform better than using a constant for every batch. Combing this with the observation that removing episodic curiosity will severely impact the performance, we believe it is safe to say (from an empirical perspective) that the learned lifelong curiosity is not critical to the performance.
> > > > > > >
> > > > > > > > (b) I think you should be careful about calling this "exploration performance" because the y-axis of Figure 4 is return, and these two quantities shouldn’t be conflated. It might be more accurate to say that the lifelong intrinsic reward does not consistently contribute positively to performance.
> > > > > > >
> > > > > > > Thank the reviewer for the suggestion. We will rephrase it as suggested.
> > > > > > >
> > > > > > > > Looking at Figure 4, in some combinations, the difference in the learning curves is actually pretty substantial (e.g. BeBold + ep_count) suggesting that the intrinsic reward results in very different behaviour than its permutations.
> > > > > > >
> > > > > > > We guess the reviewer is referring to Figure 7. We also notice this big difference between BeBold + ep_count and its permutation, and agree that the learned lifelong intrinsic reward has a non-trivial effect on the agent's behavior in this case. We will explicitly note this observation in our paper, though we believe it does not affect our findings that the learned lifelong curiosity is not critical to the performance.
> > > > > > >
> > > > > > > > Fig. 5 concerns
> > > > > > >
> > > > > > > We thank the reviewer for the constructive feedback and will remove Fig. 5 from our paper.
> > > > > > >
> > > > > > > > Calling the lifelong intrinsic rewards "learned" is misleading; they are computed from other learned quantities, unless I'm mistaken, so the word "computed" might be a good substitute.
> > > > > > >
> > > > > > > Thank the reviewer for this good suggestion. We will replace it in the final version.
> > > > > > >
> > > > > > > > I don’t understand what you mean by “Then for transitions in each trajectory, we calculate the learned lifelong intrinsic reward using models trained with r^{lifelong} only”. The only objects in your paper that could be called models and that take reward as input are RL agents.
> > > > > > >
> > > > > > > The models here refer to the networks that are used to compute the lifelong intrinsic reward (e.g., forward/inverse dynamic models in RIDE/ICM, random target predictors in RND/BeBold). We will make it clear in the final version.

---

> > > > > > > ### Author Response · Authors · 2022-12-10
> > > > > > > **Have our revision and response addressed the concerns?**
> > > > > > >
> > > > > > > Dear Reviewer GnK2,
> > > > > > >
> > > > > > > Thank you very much for the suggestions and comments during the long discussion. They have greatly improved the paper. Since the discussion period is closing soon, we are wondering if your concerns have been addressed and if the paper is in acceptable form now. If not, we would be happy to continue the discussion and/or revise the paper.

---

> > > > > > ### Comment · Reviewer_GnK2 · 2022-11-24
> > > > > > **Even more editing**
> > > > > >
> > > > > > - "rolling out the learned policy" (p. 7) "rolling out the learned policies" (you do this for every lifelong reward, yes?)
> > > > > > - "One remedy is using" (p. 9) Just "using procedurally generate environments" could mean running already-designed, poorly-generalizing, overfitting algorithms on procedurally generated environments, which wouldn't do anything other than perform poorly. I think the point you're trying to make is that designers of algorithms are likely to come up with better-generalizing new algorithms with less tendency to overfit if they test new algorithms on procedurally-generated environments, right? Your sentence doesn't say that, so you may want to rephrase it.
> > > > > > - "episodic curiosity itself" (p. 9) → I recommend "episodic curiosity alone"
> > > > > > - "the inability of lifelong curiosity" (p. 9) → "inability" doesn't stand well on its own (I find myself asking "inability to what?") so you might do better with something like "ineffectualness" or "poor performance."
> > > > > > - "roots in its ineffectiveness on" (p. 9) → Here's a good place for 'inability'! "is rooted in its inability to"
> > > > > > - "further investigations." (p. 9) ← I recommend using the uncountable form, "further investigation."
> > > > > > - "Impala: Scalable distributed deep-rl" (p. 10) ← "IMPALA: Scalable distributed deep-RL"
> > > > > > - "Figrue" (p. 15) → "Figure"

---

> > > > > > > ### Author Response · Authors · 2022-11-26
> > > > > > > **Thank you for the editing!**
> > > > > > >
> > > > > > > Thank you very much for the help. We have fixed these issues in our paper.

---

### Official Review · Reviewer_3Vou · 2022-11-01

**Confidence:** 3
**Correctness:** 2
**Technical Novelty And Significance:** 1
**Empirical Novelty And Significance:** 2
**Recommendation:** 5

**Clarity, Quality, Novelty And Reproducibility:**

The paper is easy to read. The graphics are to the point, nice and clean. Code and details are provided.

**Strength And Weaknesses:**

Strengths:
 - relevant survey
 - several ablations and analysis performed

Weaknesses:
 - the policy and value function seem to be non-recurrent. The Minigrid is a POMDP environment, so some form of memory is required. This might explain why episodic returns are so successful here.
The RND reward, for instance, should explicitly make the policy find all states in the state-space. The fact that it does not, shows that the policy cannot distinguish the states (or there is a bug somewhere else).
 - The statements are too strong: 1. There are only a few fixed environments used (really procedurally-generated is e.g. in procgen where every level is different)
 - The ablation with the randomization of the lifelong reward hints to me that there is either a bug or that, without memory, there is no to determine the reward properly from the observation alone.

Details:
- Intro p1: "gradually discourages": from my understanding the intrinsic rewards are positive, so they are not discouraging but instead are encouraging the visitation of certain states (as long as "intrinsically interesting") and then the reward for these states goes eventually to 0 (no extra incentive to go there)
- contribution 3: this finding would be ... why conjuctive?
- EQ 8: the denominator needs an additional constant to be not 1/0 (or is it post-factum computed, such that all counts are at least 1?)


**Summary Of The Paper:**

The paper evaluates several intrinsic rewards (lifelong and episodic) on some of the difficult Minigrid environments.
The results are that simple episodic rewards are better than the established lifelong once in these environments.


**Summary Of The Review:**

The paper is interesting and studies a relevant aspect of RL.
However, I have the suspicion that the paper is flawed and that the results are due to the lack of memory in the architectures.

Post-rebuttal:
Thanks for the clarification, see my answer below. I increased my score to 5.

---

> ### Author Response · Authors · 2022-11-10
> **Authors' response**
>
> Thank the reviewer for the constructive feedback. We are happy to see the comments that our paper is easy to read, and that the graphics are nice and clean. Below we respond to the comments. We also upload a revised version of the paper that incorporates the suggested changes.
>
> > `> Comment 1 (C1):` the policy and value function seem to be non-recurrent. The Minigrid is a POMDP environment, so some form of memory is required. This might explain why episodic returns are so successful here. The RND reward, for instance, should explicitly make the policy find all states in the state-space. The fact that it does not, shows that the policy cannot distinguish the states (or there is a bug somewhere else).
>
> `> Response 1 (R1):` We agree that in principle the use of memory is beneficial in POMDP environments. But in practice, the benefits are not always guaranteed. We believe our discoveries are not due to the lack of memory. First, AGAC and RAPID also use non-recurrent architecture but their performance is comparable to or even better than RIDE (which uses recurrent networks). If the memory is really critical, then they should perform much worse than RIDE. Second, as shown in BeBold paper (their Fig. 4), the authors use recurrent architectures but still find RND cannot explore beyond a small number of rooms.
>
> > `> C2:` The statements are too strong: 1. There are only a few fixed environments used (really procedurally-generated is e.g. in procgen where every level is different)
>
> `> R2:` We would like to clarify that, despite its simplicity, MiniGrid is also a procedurally-generated environment where each episode has different layouts (as mentioned in prior works such as RIDE and BeBold).
>
> > `> C3:` The ablation with the randomization of the lifelong reward hints to me that there is either a bug or that, without memory, there is no to determine the reward properly from the observation alone.
>
> `> R3:` We can confirm that our code runs without bugs. We would like to point out that AGAC and RAPID also perform pretty well without memory.
>
> > `> C4:` Intro p1: "gradually discourages": from my understanding the intrinsic rewards are positive, so they are not discouraging but instead are encouraging the visitation of certain states (as long as "intrinsically interesting") and then the reward for these states goes eventually to 0 (no extra incentive to go there)
>
> `> R4:` Thanks for this helpful comment. We have changed that sentence to "The lifelong curiosity encourages visits to the novel states that are less frequently experienced in the entire past, while the episodic curiosity encourages the agent to visit states that are relatively novel within an episode." in the revised version.
>
> > `> C5:` contribution 3: this finding would be ... why conjuctive?
>
> `> R5:` Thank the reviewer for pointing it out. We assume the reviewer means "why subjunctive?". We change it to "should" in the revised version, as suggested by another reviewer.
>
> > `> C6:` EQ 8: the denominator needs an additional constant to be not 1/0 (or is it post-factum computed, such that all counts are at least 1?)
>
> `> R6:` Yes, it is post-factum computed and all counts are at least 1. Thank the reviewer for bringing it up. We have added some clarifications in the revised version.

---

> > ### Comment · Reviewer_3Vou · 2022-11-19
> > **Rebuttal Answer**
> >
> > Thanks a lot for the clarifications and for running additional experiments to support that your results are not due to the lack of memory.
> >
> > I am still a bit unsatisfied by the amount of evaluation/variety of environments, but I appreciate that some statements were toned down in the current version.
> >
> > I understand that alternative evaluations are not straight forward. Procgen has sparse reward tasks and would have been a great testbed,l though. In the current form, the paper has value to the community even with the limited empirical scope, thus I will increase my score to 5.

---

> > > ### Author Response · Authors · 2022-11-19
> > > **Thank you for the reply**
> > >
> > > We thank the reviewer for increasing the score. We will try our best to run some experiments on the non-grid environments suggested by Reviewer bLVL.

---

> ### Author Response · Authors · 2022-11-18
> **Updated experiments results regarding recurrent networks**
>
> We run the open-sourced code of NovelD (i.e., BeBold), which uses a recurrent policy and value function. As Fig. 23 shows, NovelD hardly makes any progress when episodic curiosity is removed. This corroborates that our discoveries are not due to the lack of memory.

---

### Official Review · Reviewer_pHUd · 2022-11-02

**Confidence:** 3
**Correctness:** 3
**Technical Novelty And Significance:** 1
**Empirical Novelty And Significance:** 3
**Recommendation:** 8

**Clarity, Quality, Novelty And Reproducibility:**

The paper was very well written. Each section was presented in a clear and concise way. The authors did a good job of getting original source code from prior work, making the reproducibility of their work very high. Hyperparameter selections were well documented as well as the experimental procedure. The results appear to show a novel discovery with respect to the effects of lifelong and episodic curiosity within MiniGrid.
Minor typo in Section 3.2, the word "achieves" is misspelled in the sentence "For the goal-reaching task, the extrinsic reward ..."

**Strength And Weaknesses:**

Strengths:
* The authors seem to have done a good job of ablating curiosity configurations from recent work in this space, really trying to tease apart the contributions of both lifelong and episodic curiosity used in this prior work, particularly with respect to showing all combinations of configurations across all of the tasks they evaluated with.
* The authors do a good job of obtaining original source code from previous works, helping to eliminate potential sources of variation within their study.
* The authors presented their work in a clear and concise way that was easy to read and understand.

Weaknesses:
* The authors tend to use procedurally generated as a proxy for hard exploration task. I think this can cause some confusion with respect to their claims. I would prefer them to either be more explicit that their discovery does not necessarily apply to all procedurally generated environments, and rather to hard exploration tasks.
* I am a bit concerned that it appears as though there is little to no variance in the performance of some of the runs, even over 5 seeds. Perhaps I am missing something in these graphs?
* I would have liked to see more than 5 seeds run for a paper where the main contribution is empirical results.
* I would like to have seen mentioned or analysis the sensitivity to hyper parameter selection from the various tasks they trained on, even when those selections came from other works.


**Summary Of The Paper:**

This paper explores the how two types of curiosity, lifelong and episodic, contribute to performance and exploration when training an agent using PPO within three different hard exploration tasks from the MiniGrid evironment: KeyCorridor, MultiRoom, and ObstructedMaze. The authors claim that previous work in this setting typically involve designing for lifelong curiosity and leaving episodic curiosity as a minor complement. The authors proceed to show that the contribution of lifelong curiosity in minimal or non-existent when combined with episodic curiosity and as such does not improve performance of the agents within these tasks.  The authors then continue to demonstrate that the reason for this is likely due to lifelong curiosity not assigning higher value to novel states.

**Summary Of The Review:**

Overall, I would recommend acceptance of this paper. I found the work to be a good empirical study with results that have the potential to shape future work in the space of lifelong curiosity or other intrinsic reward design. I do think that it would do well for the authors to make clear that rocedurally generated actually maps to hard exploration task, since the former is a bit vague in definition.

---

> ### Author Response · Authors · 2022-11-10
> **Authors' response**
>
> Thank the reviewer for the constructive feedback. We are thrilled to see the comments that our work has done a good job of ablating curiosity configurations from recent work, and that our paper is clearly and concisely presented and easy to read. Below we respond to the comments. We also upload a revised version of the paper that incorporates the suggested changes.
>
> > `> Comment 1 (C1):` The authors tend to use procedurally generated as a proxy for hard exploration task. I think this can cause some confusion with respect to their claims. I would prefer them to either be more explicit that their discovery does not necessarily apply to all procedurally generated environments, and rather to hard exploration tasks.
>
> `> Response 1 (R1):` Thank the reviewer for the suggestion. We have revised the conclusion section to make it more explicit that our discovery does not necessarily apply to all procedurally generated environments.
>
> > `> C2:` I am a bit concerned that it appears as though there is little to no variance in the performance of some of the runs, even over 5 seeds. Perhaps I am missing something in these graphs?
>
> `> R2:` The reason for this effect might be that we use a moving average strategy (as done in RIDE) to smooth the curves. Taking the "ep_visit only" result on `KeyCorridor-S6R3` as an example, we re-plot it without smoothing (see this [anonymized figure](https://i.imgur.com/CP0lqCt.png)). We can see there is variance across different runs. Nevertheless, we will make it clear in the final version to reduce confusion.
>
> > `> C3:` I would have liked to see more than 5 seeds run for a paper where the main contribution is empirical results.
>
> `> R3:` Thanks for the suggestion. We are re-running the experiments for 10 runs and will update the results soon.
>
> > `> C4:` I would like to have seen mentioned or analysis the sensitivity to hyper parameter selection from the various tasks they trained on, even when those selections came from other works.
>
> `> R4:` We conduct an ablation study on a critical hyper-parameter: the intrinsic reward coefficient $\beta$. The best searched value of $\beta$ for each setup is summarized in Appx. A.5. We show an example in Fig. 19 about the influence of $\beta$ on performance. We will make all results publicly available and can also upload them to the supplementary material if needed. For PPO hyperparameters, we simply follow a common configuration in https://github.com/openai/baselines with little tuning for the learning rate in pilot experiments. For the loss coefficients of forward/inverse models and RND, we follow the default setting in RIDE's code. We are willing to add more analysis if there are other things the reviewer expects.
>
> > `> C5:` Minor typo in Section 3.2, the word "achieves" is misspelled in the sentence "For the goal-reaching task, the extrinsic reward ..."
>
> `> R5:` Thank the reviewer for pointing it out. We have fixed it in the revised version.

---

> > ### Comment · Reviewer_pHUd · 2022-11-18
> > **Thank You**
> >
> > Thank you for addressing my concerns. I appreciate that you took the time to double your seeds for your experiments and cleared up some other confusion I had.
> >
> > My only concern left is part of a similar thread that the other reviewers have touched on in different ways is that procedurally generated environments still appear in the text as something more generic and only until the conclusion is it addressed that this might not hold. I would like to have seen the initial claim be something more explicit and closer to:
> >
> > "We conduct a comprehensive study on the lifelong and episodic curiosity in exploration and
> > find that the episodic curiosity overlooked by previous works is actually the more important
> > ingredient for efficient exploration in procedurally-generated **grid worlds**"
> >
> > It is a minor nit however since it is possible that the other researchers in this area use the term loosely as well.

---

> > > ### Author Response · Authors · 2022-11-18
> > > **Thank you for the suggestion**
> > >
> > > We thank the reviewer for the suggestion. We have revised our paper to explicitly mention "gridworld" in the contribution part and several other places in the paper.

---

> ### Author Response · Authors · 2022-11-12
> **Updated results with 10 runs**
>
> We rerun the main experiments (Fig. 2) for 10 runs on `KeyCorridor-S4R3` and include the results in Fig. 21. The performance are close to those in Fig. 2. Experiments for other environments are still running. We will update the results as soon as the experiments are finished.

---

### Author Response · Authors · 2022-11-18
**Summary of the additional experiments**

We thank the reviewers again for their constructive and very helpful comments. Below we give a summary of the additional experiments we have run (or are running) in the discussion period.

* Run our experiments with more seeds:

  We increase the number of runs from 5 to 10 and include the results in Fig. 21. Currently, the experiments on `KeyCorridor-S4R3` and `KeyCorridor-S6R3` have finished. We will update the results on other tasks as soon as possible in an anonymized link during discussion stage 2.
* Run NovelD's open-sourced code:

  We have re-run the open-sourced code of NovelD on two tasks that are not reported in their paper (`MultiRoom-N10S10` and `MultiRoom-N12S10`), and include the results in Fig. 23 in the revised version. In addition, we also conduct an ablation experiment of removing the episodic curiosity in NovelD. The results (also in Fig. 23) indicate that our discoveries are not due to using non-recurrent networks, which should resolve Reviewer 3Vou's major concerns.
* Compare with lifelong count-based exploration methods such as $\frac{1}{\sqrt{N_{life}(s_{t+1})}}$:

  We conduct the experiments as suggested, and include the results in Fig. 22. Currently, the experiments on `KeyCorridor-S4R3` and `KeyCorridor-S6R3` have finished. We will update the results on other tasks as soon as possible in an anonymized link during discussion stage 2.

We will be happy to answer further questions and conduct other necessary experiments during discussion stage 2.

---

### Author Response · Authors · 2022-12-10
**Updates on experiments**

We thank all the reviewers for their valuable feedback, which greatly helped improve our paper. Here we update the experiment results and provide a summary of the experiments we ran during the discussion period. Since the discussion period is closing soon, we are wondering if the concerns have been addressed. We would be happy to continue the discussion and/or revise the paper.

* Run our experiments with more seeds:

  We increase the number of runs from 5 to 10 and update the results in this [anonymized figure](https://i.imgur.com/ApmuF3Y.jpg).

* Run NovelD's open-sourced code:

  We have re-run the open-sourced code of NovelD on two tasks that are not reported in their paper (`MultiRoom-N10S10` and `MultiRoom-N12S10`), and include the results in Fig. 23 in the revised version. In addition, we also conduct an ablation experiment of removing the episodic curiosity in NovelD. The results (also in Fig. 23) indicate that our discoveries are not due to using non-recurrent networks, which should resolve Reviewer 3Vou's major concerns.

* Compare with lifelong count-based exploration methods such as $\frac{1}{\sqrt{N_{life}(s_{t+1})}}$:

  We conduct the experiments as suggested, and update the results in this [anonymized figure](https://i.imgur.com/pLsLox7.png).

* Experiments on non-gridworld environments

  We made our best to run the suggested experiments, and below are the things we tried in the last 2 weeks:
    * We notice that NovelD and RIDE also use episodic counts for complex environments such as Atari and Vizdoom. Therefore, we tried to run experiments on Atari and Vizdoom with their open-sourced code. However, we find that current NovelD's code only supports MiniGrid ([see here](https://github.com/tianjunz/NovelD/blob/ed9680e9ddb81e4b46a7020f03f6c483b06c1d8d/src/algos/bebold.py#L195)). For RIDE, we first encountered some environment wrapper bugs when running their code on Atari and Vizdoom. Even though we fixed the bugs, there were NaN issues and it was hard to pin down the cause.
    * We tried to implement the reachability network [2] in our codebase but it did not work even on MiniGrid. We tried a grid search for the hyper-parameters on MultiRoom-N7S8, e.g., learning rate, number of positive and negative pairs, training epochs, intrinsic reward multiplier. But we still have not obtained good results. The average return is almost always 0.
    * We notice that the ablation study in Figure 3 of the NGU paper [1] shows that episodic intrinsic rewards contribute more to the performance than lifelong intrinsic rewards in non-gridworld environments (Atari games). We tried to reproduce their results with [a third-party re-implementation](https://github.com/michaelnny/deep_rl_zoo). The experiments are still running and we will update the results in the final version. We believe the ablation experiments in NGU paper can corroborate our findings for non-gridworld environments.


[1] Badia, Adrià Puigdomènech, et al. "Never give up: Learning directed exploration strategies." arXiv preprint arXiv:2002.06038 (2020).

[2] Savinov, Nikolay, et al. “Episodic curiosity through reachability.” ICLR. 2019.

---

> ### Comment · Reviewer_bLVL · 2022-12-11
> **Thank you and one question**
>
> I truly appreciate the authors sincerely dealing with reviewers' concerns.
>
> I would like to hear the authors' opinions about the results of the lifelong count-based exploration.
>
> The result shows lifelong count has almost the same performance across six different environments consistently.
> Correct me if I am wrong. In my opinion, this means that the problem with Minigrid is not between episodic curiosity and lifelong curiosity, but whether the actual count-based method is used or not. It converts the entire problem from **episodic curiosity vs. lifelong curiosity** (authors' argument) to **count-based curiosity vs. prediction-based curiosity**.
>
> However, I believe since existing methods combine count-based episodic curiosity with prediction-based lifelong curiosity, focusing only on that curiosity is not totally wrong. I recommend including the result of the lifelong count method in the paper and writing an analysis about it.

---

> > ### Author Response · Authors · 2022-12-11
> > **Thank you for the insightful comment!**
> >
> > Our original take on this result is that lifelong counts may behave quite similarly to episodic counts since every episode is new in procedurally-generated environments. We also agree with the reviewer that "count-based curiosity vs. prediction-based curiosity" might be the real problem here. We will include the result of the lifelong count method in the main paper and add an analysis about comparing count-based and prediction-based curiosity.

---

### Decision · Program_Chairs · 2023-01-20

**Decision:**

Accept: poster

**Justification For Why Not Higher Score:**

The paper is interesting to a subcommunity of exploration in RL which focuses on procedurally-generated environments. The paper is not necessarily interesting for the RL community thinking about exploration outside these environments, so it is not something for a large audience and thus I don’t see the need for a spotlight/oral slot; not to mention it was a borderline paper.

**Justification For Why Not Lower Score:**

The main message of the paper could make the community it is targeted to to rethink exploration. This is a good reason to accept the paper.


**Metareview: Summary, Strengths And Weaknesses:**

When solving exploration problems procedurally generated environments the field often does so through a combination of two types of bonus that are multipled together: a lifelong one, which takes into consideration the agent’s experience throughout its entire lifetime, and an episodic one, which considers only the agent’s experience in that particular episode/problem instance.

This paper performs a thorough analysis of the impact of each of these bonuses (lifelong and episodic) in the agent’s performance, and it shows that, in procedurally generated MiniGrid environments, the types of lifelong bonuses they investigated have little to no effect on the agent’s performance, while the episodic bonuses they investigated explain pretty much all the gain in performance when using these bonuses. This paper is an interesting paper to the community of exploration in procedurally generated environments because it suggests the community should maybe rethink how exploration should be done.

Importantly, this paper led to a lot of discussion between the reviewers because while they all agree about its potential to shape how the community thinks about generalization, there were also several concerns about presentation and the claims being made. The authors did address several of these concerns during the discussion phase, but I’d encourage the authors to read all reviews carefully again and make sure other comments are addressed as well. Of particular note are the concerns around Section 3.4, such as the static policy rollout experiment, the permutation experiment, and Figure 5. The authors said they will remove Figure 5 and that they will make Section 3.4 much more speculative, and I recommend they do so, while carefully following the reviewers’ suggestions.

Personally, I am concerned about how the episodic bonus leverages so much domain knowledge while the lifelong one is very general. In that sense, it is not surprising that the episodic bonus carries so much weight. I would have found more interesting an experiment where the same function is used for the lifelong and episodic bonuses, allowing us to better understand if the observed phenomenon is due to the particular choice of bonuses chosen or indeed because a reset at each episode is the more effective way of doing so.

Nevertheless, at the end we concluded that the benefits outweighs the downsides of the paper and as long as the paper addresses the reviewers concerns it will be sharing an important message to the PCG-exploration community.


**Note From Pc:**

if the above contains the word "oral" or "spotlight" please see: "oral" presentation means -> notable-top-5% and "spotlight" means -> notable-top-25%. As stated in our emails, we are disassociating presentation type from AC recommendations

**Summary Of Ac-Reviewer Meeting:**

I scheduled the meeting expecting reviewers e8Yz, bLVL, pHUd, and GnK2. Reviewer 3Vou didn’t find a slot that would work for them and the others. At the time of the meeting reviewer pHUd didn’t show up.


Reviewer e8Yz was very positive about the paper because of their personal experience in the problem and the fact that the insights the authors report had also been observed by them, validating the authors’ claims. The reviewer focused on the big picture of the main message of the paper and they supported the claim that this would be an interesting paper to that subcommunity and that could change how people thought about a problem. Every reviewer in fact agreed that this paper had an interesting message.

We then moved to discuss the main concerns, such as reviewer’s bLVL concerns about the generality of the method, which we agreed the authors did a good job at toning down their claims during the response period, and reviewer’s GnK2 concern about Section 3.4. Reviewer GnK2 hadn’t seen the latest response by the authors and the reviewer was more comfortable accepting the paper if the authors were to follow through with their promises in the last response. The other reviewers in the meeting did agree with the concerns raised by reviewer GnK2, but they also agreed that it didn’t change the main message.

At the end reviewer GnK2 said they would be comfortable with the paper being accepted if the authors made Section 3.4 more speculative; reviewer bLVL said they were on the fence at that point, and reviewer e8Yz continued to support the paper. Given the interestingness of the main message and the fact that no one raised additional red flags we agreed to recommend the acceptance of the paper.